# Who with whom: functional coordination of E2 enzymes by RING E3 ligases during poly-ubiquitylation

Christian Lips[1,†], Tobias Ritterhoff[2,†], Annika Weber[1,‡], Maria K Janowska[2], Mandy Mustroph[1], Thomas Sommer[1,3] & Rachel E Klevit[2,*]

## Abstract

**Protein modification with poly-ubiquitin chains is a crucial process involved in a myriad of cellular pathways. Chain synthesis requires two steps: substrate modification with ubiquitin (priming) followed by repetitive ubiquitin-to-ubiquitin attachment (elongation). RING-type E3 ligases catalyze both reactions in collaboration with specific priming and elongating E2 enzymes. We provide kinetic insight into poly-ubiquitylation during protein quality control by showing that priming is the rate-determining step in protein degradation as directed by the yeast ERAD RING E3 ligases, Hrd1 and Doa10. Doa10 cooperates with the dedicated priming E2, Ubc6, while both E3s use Ubc7 for elongation. Here, we provide direct evidence that Hrd1 uses Ubc7 also for priming. We found that Ubc6 has an unusually high basal activity that does not require strong stimulation from an E3. Doa10 exploits this property to pair with Ubc6 over Ubc7 during priming. Our work not only illuminates the mechanisms of specific E2/E3 interplay in ERAD, but also offers a basis to understand how RING E3s may have properties that are tailored to pair with their preferred E2s.**

**Keywords** E2 conjugating enzyme; ER-associated protein degradation; linchpin; RING E3 ligase; ubiquitin

**Subject Categories** Post-translational Modifications & Proteolysis; Structural Biology

**The EMBO Journal (2020) 39: e104863**

## Introduction

The post-translational modifier ubiquitin (Ub) controls virtually every process in eukaryotic cells. Protein modification with Ub is carried out by three sequentially acting enzymes. The E1 enzyme activates the Ub C-terminus for transfer to the active site of an E2 enzyme to form an E2~Ub thioester conjugate (in this text, "~" signifies the thioester linkage). E3 ligases are generally attributed with two functions: They bring an E2~Ub conjugate and a suitable substrate into proximity ("recruitment"), and they catalyze the transfer of Ub from the E2 to the substrate thus ensuring spatiotemporal control of the process ("stimulation"). All eukaryotic organisms from yeast to human have a hierarchy of E1/E2/E3 enzymes, with 1–2 human E1s, up to 36 E2s, and hundreds of E3s. The hierarchy dictates that a given E2 must work with numerous E3s, but it is also true that numerous E2s can work with a given E3. For the largest family of E3s, the RING E3s, the identity of the collaborating E2 defines the product of the reaction (i.e., mono- or poly-ubiquitylation) and, therefore, the biological outcome (Christensen *et al*, 2007). How an E3 pairs with a specific E2—from the set of enzymes it can physically interact with—to determine a distinct functional outcome remains an open question.

The ability to stimulate E2 activity is rooted in the way RING E3 ligases function. They bind an E2~Ub conjugate and stimulate Ub transfer to a substrate without participating directly in the reaction. In solution, Ub conjugated to an E2 is highly flexible, creating a dynamic conformational equilibrium that is associated with low transfer activity (Fig 1A; Pruneda *et al*, 2011). RING E3s work allosterically by restricting the flexible Ub toward so-called closed conformations that involve non-covalent interactions between a hydrophobic surface patch on Ub and a surface proximal to the E2 active site (Reverter & Lima, 2005; Saha *et al*, 2011; Wickliffe *et al*, 2011; Dou *et al*, 2012; Plechanovová *et al*, 2012; Pruneda *et al*, 2012; Brown *et al*, 2014; Kelly *et al*, 2014; Branigan *et al*, 2020). In general, RING E3s achieve this shift in E2~Ub conformational equilibria through a conserved residue, the allosteric linchpin, that engages both the Ub and the E2 to restrict their relative orientations. A shift toward closed conformations is associated with a dramatic increase in Ub transfer activity from an E2 (Pruneda *et al*, 2012; Branigan *et al*, 2020). Structural data reveal that a RING linchpin engages Ub and the E2 through hydrogen

1  Max Delbrück-Center for Molecular Medicine in the Helmholtz Association, Berlin-Buch, Germany
2  Department of Biochemistry, School of Medicine, University of Washington, Seattle, WA, USA
3  Lady Davies Guest Professor, Technion-Israel Institute of Technology, Haifa, Israel
   *Corresponding author. Tel: +1 206 543 5891; E-mail: klevit@u.washington.edu
   †These authors contributed equally to this work
   ‡Present address: MRC Laboratory of Molecular Biology, Cambridge, UK

bonding (Fig 1B; Dou *et al*, 2012; Plechanovová *et al*, 2012; Pruneda *et al*, 2012; Branigan *et al*, 2015). In a survey of yeast and human RING-type E3s, arginine, with its multiple hydrogen bond donor groups, is most commonly found at the linchpin position (Figs 1C and EV1 for yeast and human, respectively). Surprising, at least half of all human and yeast RINGs feature a residue other than arginine at the linchpin position. Some have been reported as functional, but less efficient linchpins than arginine (Yin *et al*, 2009; Pruneda *et al*, 2012; Scott *et al*, 2014; Stewart *et al*, 2017); some are residues that can potentially act as a hydrogen bond donor, but have not been experimentally verified as functional linchpins; some are residues that lack hydrogen bond potential altogether. Prominent examples include yeast RING E3s Rad16 (histidine) and Rad18 (leucine), both involved in DNA damage pathways, and Rbx1 (asparagine), the common RING module of the large family of cullin-RING ligases (CRLs). How these RINGs stimulate the ubiquitylation activity of their paired E2s is largely unknown.

The best-understood types of protein ubiquitylation involve the formation of poly-Ub chains of specific linkages (e.g., Ub K48- or K11-linked chains serve as signals for proteasomal degradation) (Komander & Rape, 2012; Metzger *et al*, 2013b). For RING-type E3s, substrate modification with a poly-Ub chain involves two types of Ub transfer reactions: (i) a "priming" step in which a single Ub is transferred to a protein substrate and (ii) "elongation" reactions in which Ub is attached to the substrate-modified Ub to generate a poly-Ub chain (reviewed in Stewart *et al*, 2016; Deol *et al*, 2019). The priming step is biochemically diverse due to the varied nature of potential substrates and Ub attachment sites, whereas chain elongation entails highly specific and repetitive reactions. The different biochemical requirements of priming and elongation have led to the recognition of dedicated E2s for each step. Dedicated chain-elongating E2s that specialize in the synthesis of specific Ub linkages such as K11-, K48-, and K63-linked chains are well known. However, the identities and mechanisms of specific mono-ubiquitylating priming E2s and of E2s that appear to carry out both functions are less well understood. For example, members of the UbcH5 (UBE2D) family are highly promiscuous *in vitro* and potentially function in both priming and elongation capacities. It is likely that their functional action, i.e., priming or elongation, is dictated by the particular E3 with which such E2s cooperate in a given case (summed up by Stewart *et al*, 2016), but how such a selection is achieved is not known.

Examples of RING E3s that are confirmed to collaborate with separate priming and elongation E2s to catalyze attachment of poly-Ub chains to substrates continue to be reported (Rodrigo-Brenni & Morgan, 2007; Saha & Deshaies, 2008; Kleiger *et al*, 2009; Parker & Ulrich, 2009; Pierce *et al*, 2009; Williamson *et al*, 2009; Wu *et al*, 2010a; Kelly *et al*, 2014; Scott *et al*, 2014, 2016; Dove *et al*, 2016; Weber *et al*, 2016; Hill *et al*, 2019). Still, our understanding of how E3 ligases control the interplay of E2 enzymes during the two steps of poly-ubiquitylation is rather limited. To understand how RING E3s utilize E2s for priming and elongation reactions, we focused on a well-defined molecular poly-ubiquitylation system. ER-associated protein degradation (ERAD) is a highly conserved protein quality control pathway that targets misfolded ER-resident proteins and marks them for proteasomal degradation (Christianson & Ye, 2014). Yeast ERAD relies on two RING E3 ligases, Hrd1 and Doa10, and two E2 enzymes, Ubc6 and Ubc7, to modify substrates with

K48-linked Ub chains (Fig 1D). Ubc6 harbors a C-terminal transmembrane domain and is thought to function primarily as a mono-ubiquitylating priming E2 (Weber *et al*, 2016). Ubc7 is recruited to the membrane by the accessory protein Cue1; binding to the Ubc7-binding region (U7BR) of Cue1 renders Ubc7 competent for Ub transfer, while the Cue1 domain of the protein serves to align a growing poly-Ub chain for K48-specific elongation by Ubc7 (Kostova *et al*, 2009; Bagola *et al*, 2013; Metzger *et al*, 2013a; von Delbrück *et al*, 2016). The E3 Doa10 employs Ubc6 and Ubc7 for priming and elongation, respectively (Weber *et al*, 2016), while Hrd1 relies predominantly on Ubc7 (Bays *et al*, 2001) (Fig 1E). Hrd1-targeted ubiquitylation is largely unaffected by the absence of Ubc6, implying that Ubc7 can carry out both priming and elongation with Hrd1. Nevertheless, no direct evidence for Ubc7's priming activity has been reported to date. Notably, Doa10 features a potentially suboptimal histidine as its linchpin, while Hrd1 harbors a canonical arginine (Fig 1F). This feature implies different requirements for E3-mediated E2 stimulation in the two RING E3s, but this notion has not been directly addressed experimentally.

Our comparisons of the two non-redundant ERAD E3 ligases and their interacting E2 enzymes lead to several key observations and conclusions. First, we present direct *in vitro* evidence that chain-elongating Ubc7 can work as a priming E2. Second, E3-mediated E2 stimulation contributes to *in vivo* substrate degradation rates in the priming step, but is dispensable in the chain elongation step. Third, Hrd1 and Doa10 exploit different modes of E2 stimulation that are tailored to the properties of their preferred priming E2. Hrd1 has a high affinity for Ubc7 and relies on its canonical allosteric linchpin to stimulate the E2 in the priming reaction. Ubc6 has high basal activity and does not absolutely rely on stimulation by a functional linchpin for RING-mediated ubiquitylation. This enables Doa10 with its non-canonical linchpin residue to use Ubc6 and not Ubc7 for priming, while not affecting the E3's use of Ubc7 for chain elongation to a degree that impinges on protein degradation rates. Altogether, this study provides insight into how E3s functionally pair with their potential E2s to enable specific functional outcomes.

# Results

### Hrd1 selects Ubc7 through high binding affinity

To understand the E2 preferences of Hrd1 and Doa10, we characterized the catalytically relevant combinations of each E3 ligase with Ub conjugates of either E2. Isothermal titration calorimetry (ITC) was performed using the E3 RING domains, a C-terminally truncated version of Ubc6, and Ubc7 in complex with the U7BR of Cue1, which renders the E2 competent for Ub transfer (Bagola *et al*, 2013). As the native Ubc6~Ub thioester is susceptible to hydrolysis (see also later Fig 5C), we engineered stable mimics of E2~Ub thioester conjugates by generating a covalent disulfide linkage between a Ub(G76C) mutant and the E2 active site cysteine [(Lorenz *et al*, 2016); "-SS-" signifies the disulfide linkage for these conjugates].

The Hrd1 RING binds the U7BR/Ubc7-SS-Ub conjugate more than one order of magnitude stronger than it binds the Ubc6-SS-Ub conjugate, with a $K_D$ in the low µM range (Fig 2A). This implies that Hrd1 selects between Ubc6 and Ubc7 on the basis of E2~Ub/E3 affinity, consistent with the reported Ubc6 independence of Hrd1

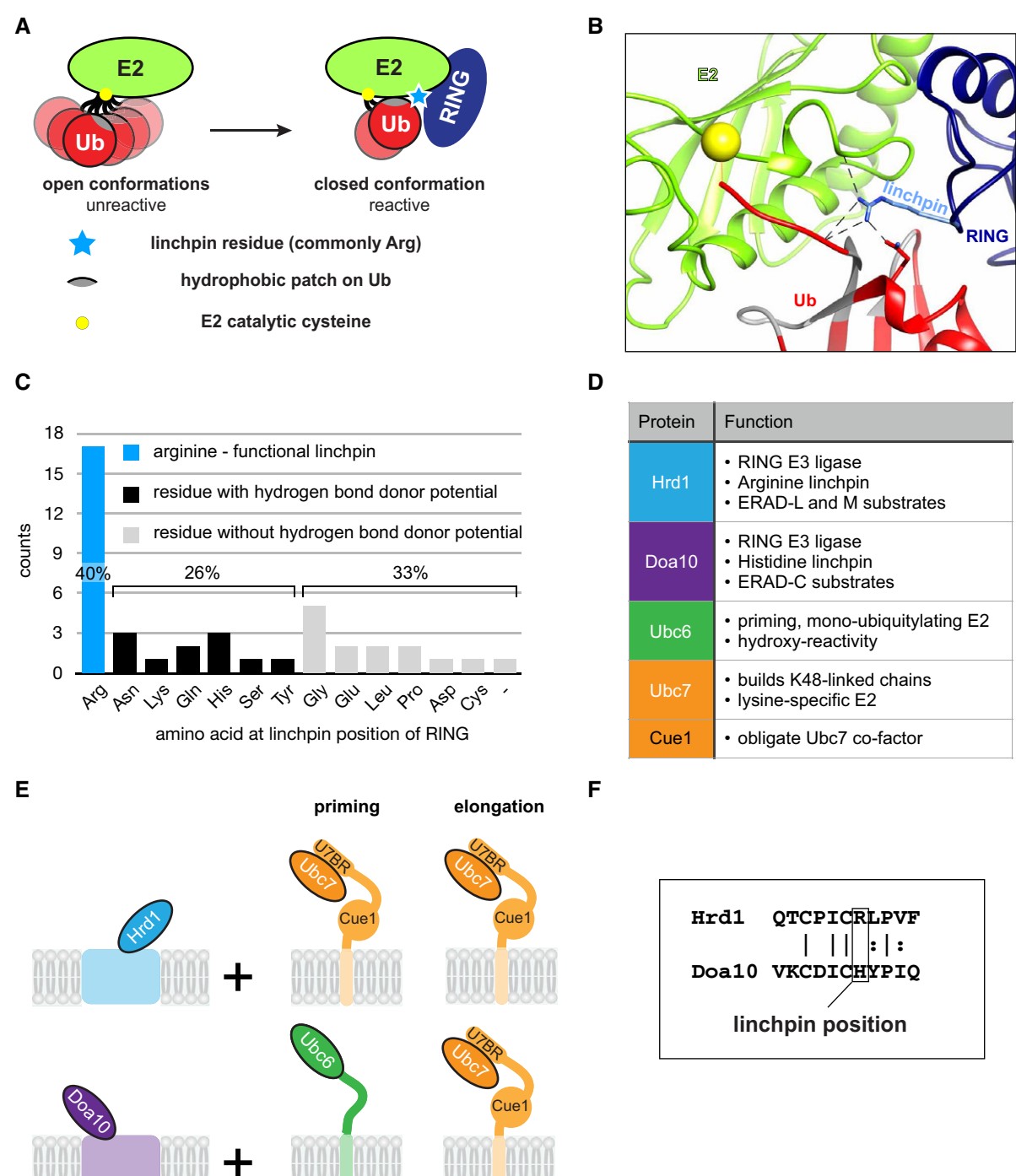

**Figure 1. Ubiquitylating enzymes in yeast ERAD and mechanism of RING-mediated E2 stimulation.**

A  Current model of RING E3-mediated E2 activation.

B  RING linchpin-mediated interactions in an E2~Ub/RING complex. Crystal structure of UbcH5a-Ub/RNF4 (PDB: 4AP4) highlighting the hydrogen bond interactions involving the RING linchpin arginine to side chain and backbone atoms on both the E2 and Ub.

C  Histogram of amino acid frequencies at the linchpin position of all 42 yeast RING domains. The linchpin position was defined as the residue at the $n + 1$ position after the final $Zn^{2+}$ ligand coordinating residues. "–" refers to the atypical RING of Pib1, which lacks the loop containing the linchpin. The SP-RING domains of the two yeast SUMO E3 ligases Siz1 and Siz2 as well as the RING1 domains of the two yeast RBR E3 ligases Hel1 and Itt1 were excluded from this analysis.

D  Summary of the relevant properties of E2 enzymes and E3 ligases of the yeast ERAD system.

E  Cartoon showing ERAD E2 pairings for both E3 ligases during priming and elongation (see text for details).

F  Sequence alignment of Hrd1 and Doa10 for the final $Zn^{2+}$-binding loops (C-X-X-C) showing the linchpin position.

substrate degradation (Bays *et al*, 2001). In contrast, the Doa10 RING binds each E2 conjugate with similar weak affinity, with $K_D$s in the high μM range, consistent with this E3 cooperating with both E2s. We note that the dissociation constants measured for the soluble versions of the RING domains and E2 conjugates reflect the intrinsic affinity for these functional protein–protein interactions and, therefore, the probability of an E3 engaging a particular E2~Ub. Tethering of the relevant components in the ER membrane (see Fig 1E) may overcome the relatively low intrinsic affinity by providing high local concentrations of an E3 and its E2~Ub conjugates.

Hrd1 is presumed to collaborate with Ubc7 for both priming and chain elongation steps, but direct evidence of substrate priming by Ubc7 has not been reported. We implemented an *in vitro* substrate ubiquitylation assay in which a sequence derived from bovine RNase A ("S-peptide") is genetically fused to an E3 (e.g., "S-Hrd1") to serve as a high-affinity substrate recruiter for an RNase A variant that lacks its S-peptide sequence (referred to here as "RNase"; Bays *et al*, 2001). The Ub(K48R) mutant was used in reactions containing Ubc7 to limit reactions to the priming step by preventing K48-linked poly-Ub chain formation. Reactions that contain a functional E2/E3 pair will generate a product that corresponds to RNase modified with Ub. The pairing of Ubc7 and S-Hrd1 yields a robust RNase-Ub band and a fainter RNase-Ub$_2$ band, which likely reflects double mono-ubiquitylated RNase (Fig 2B). In the absence of S-Hrd1, Ubc7-mediated ubiquitylation is negligible. Importantly, this result provides direct evidence that the efficient chain-elongating E2 Ubc7 is able to prime a substrate. The pairing of Ubc6 and S-Hrd1 also generated modified RNase, albeit to a lesser extent, and this reaction does not depend entirely on the presence of S-Hrd1. Importantly, the stimulatory effect of S-Hrd1, i.e., the difference of RNase modification in the presence of the E3 compared to its absence, is much greater in the case of Ubc7 than Ubc6. Altogether, the results indicate that Ubc6/S-Hrd1 can, in principle, form a functional E2/E3 pair for priming, but is much less efficient than the Ubc7/S-Hrd1 pair.

Although the *in vitro* ubiquitylation assay provided insights into the capability of Ubc6 and Ubc7 to prime a substrate, RNase is an artificial substrate that interacts with the E3 ligase in a non-native way. To obtain a quantitative measure of E2 activity and the potential of E3s to stimulate it without confounding contributions from a substrate, we monitored the kinetics of Ub transfer from preformed E2~Ub conjugates to a small nucleophile (see Fig EV2A; Pickart & Rose, 1985; Wenzel *et al*, 2011). Nucleophile is present at huge molar excess thus circumventing the need for a specific substrate interaction. Ub discharge from Ubc6~Ub and U7BR/Ubc7~Ub was followed as a function of time (Fig 2C), and rates were extracted (Fig 2D), providing a metric for substrate-independent E2 discharge activity. While U7BR/Ubc7~Ub undergoes aminolysis like most E2s, Ubc6 has been reported to be hydroxy-reactive (Wang *et al*, 2009; Weber *et al*, 2016). We therefore used ethanolamine as the nucleophile, as it provides both an amino and a hydroxy group. While this experimental design allows for uniform reaction conditions, a direct comparison of the E2s' discharge rates is inadvisable due to the inherent reactivity profile differences. Also, the discharge rates reported here are not directly transferrable to ubiquitylation of substrate *in vivo*. Despite these caveats, the assays allow assessment of the stimulatory effect of an E3 for a given E2, which we report as the ratio of the discharge rates with and without E3 (Fig 2E). U7BR/

Ubc7~Ub discharges very slowly in the absence of an E3 and Hrd1 greatly accelerates the process. In contrast, Ubc6~Ub discharges quite rapidly on its own and is only mildly stimulated by Hrd1. Quantitatively, Hrd1 is roughly an order of magnitude more potent at stimulating discharge from U7BR/Ubc7 than from Ubc6. These results are consistent with the observations made in the *in vitro* substrate ubiquitylation assays and in line with the difference in binding affinity. They suggest that the difference observed in the *in vitro* ubiquitylation assay (see Fig 2B) is due to Hrd1's more effective stimulation of Ubc7. Altogether, the binding and functional assays indicate that the known functional *in vivo* preference of Hrd1 for Ubc7 over Ubc6 for priming is dictated mainly by binding affinity.

### RING linchpin plays a key role in Ubc7 stimulation

Having established that Ubc7 can carry out priming with Hrd1, we wondered why it reportedly does not do so with Doa10, especially as the competing E2 for priming, Ubc6, binds with comparably low affinity to Doa10 (see Fig 2A). Hrd1 and Doa10 differ in their linchpin residues (Arg and His, respectively; see Fig 1F), suggesting the answer might lie in how the two E3s stimulate Ubc7. To investigate the contribution of linchpins in the stimulation of Ubc7, we generated Hrd1 and Doa10 RING variants that harbor one of four amino acids at their linchpin positions (arginine—potent hydrogen bond donor and linchpin of Hrd1; histidine—potential hydrogen bond donor and linchpin of Doa10; alanine—no hydrogen bond potential; or glutamate—hydrogen bond acceptor and charge reversal of arginine).

A strong dependence on linchpin identity was observed in Ub discharge assays for U7BR/Ubc7 acting with either E3 ligase (Fig 3A): For both, arginine is most effective, histidine and alanine are much less effective, and glutamate provides only minor stimulation of U7BR/Ubc7 activity over a "no E3" control. Wild-type Hrd1 (400R) elicits the highest Ub discharge rates from U7BR/Ubc7 with rates achieved by wild-type Doa10(94H) being roughly three times lower. Notably, Doa10 with an arginine linchpin (94R) is as effective as wild-type Hrd1. These observations imply that Hrd1-mediated Ubc7 stimulation is driven by the identity of the linchpin.

### The mono-Ub to di-Ub step is the slowest of Ub chain elongation, but not rate-determining for protein degradation

Doa10's suboptimal histidine linchpin and Ubc7's strong linchpin dependence offer a rationale for why this E3 does not rely on Ubc7 for priming. How this affects Doa10's preferred use of Ubc6 for priming will be discussed later. However, Doa10 does engage Ubc7 for chain elongation, begging the question how this is carried out. We implemented an *in vitro* assay in which one Ubc7-dependent elongation step can be monitored at a time (Fig EV3A; von Delbrück *et al*, 2016). The first three reactions of chain elongation were recapitulated by adding Alexa 488-labeled Ub to mono-Ub, di-Ub, or tri-Ub to generate di-Ub, tri-Ub, or tetra-Ub, respectively. Reactions were carried out in the absence or presence of Hrd1 or Doa10 or their linchpin variants, respectively, as well as a variant of Cue1 that contains both the U7BR and the Cue1 domain. Reaction rates for each step were derived from quantification of the time courses (Fig 3B).

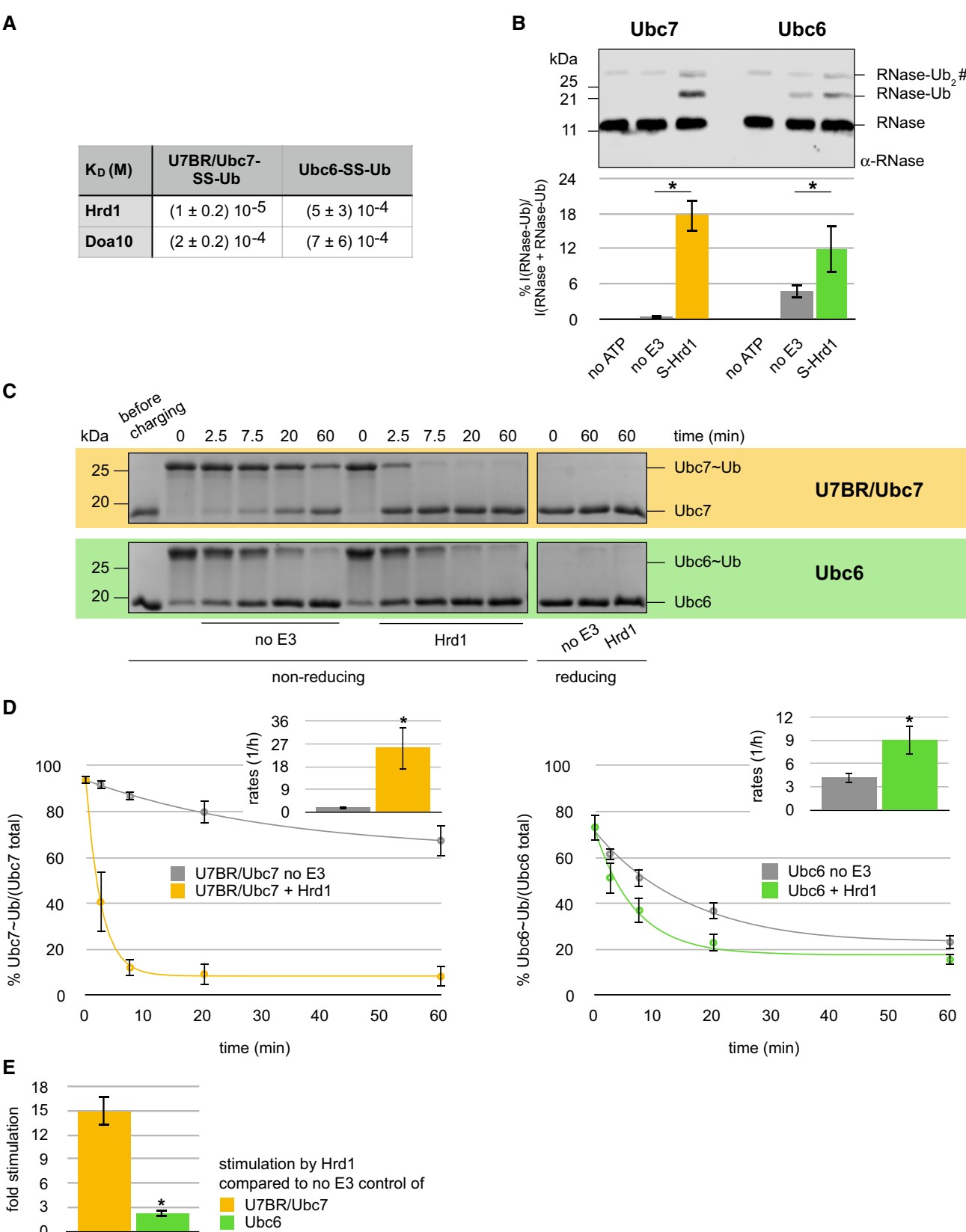

Figure 2.

**Figure 2. Hrd1 selects Ubc7 through high binding affinity.**

A  Dissociation constants for ERAD E2-SS-Ub/E3 pairs. $K_D$ values were determined by ITC titration of Hrd1 or Doa10 to Ubc6-SS-Ub and U7BR/Ubc7-SS-Ub, respectively. Due to the weak binding, errors are high, so the constants are reported to a single significant figure.

B  *In vitro* substrate ubiquitylation assay for Hrd1 with Ubc7 (left) and Ubc6 (right). Ubc7 reactions contained equimolar amounts of Cue1 and were performed with Ub(K48R). Top: representative immunoblot using a poly-clonal α-RNase A antibody; "no E3" reactions do not contain S-Hrd1, "no ATP" reactions do not contain ATP, but S-Hrd1. The RNase-Ub$_2$ band co-migrates with a non-specific band (#) common to all samples. Bottom: Quantification of RNase-Ub signals is shown. Values are reported as means ± standard deviation ($n = 3$). Significances for pairwise comparisons were determined by one-way ANOVA test; *$P < 0.05$. For clarity, only significances related to the "no E3" control of a given E2 are shown.

C  *In vitro* Ub nucleophile discharge assays for Hrd1 with U7BR/Ubc7 (top—yellow) and Ubc6 (bottom—green) with ethanolamine as nucleophile. Representative Coomassie gels are shown.

D  Quantification of Ub nucleophile discharge assays for Hrd1 with U7BR/Ubc7 (left) and Ubc6 (right). Plots of E2~Ub discharge (dots) as a function of time with first-order reaction models fitted to the discharge data (lines) are shown. Values for each time point are reported as means ± standard deviation ($n = 3$). Insets show reaction rates derived from these fits. Significances were determined by Student's *t*-test; *$P < 0.05$.

E  Stimulation of U7BR/Ubc7 and Ubc6 discharge activities by Hrd1. E2 stimulation is reported as the ratio of rates derived from RING-catalyzed reactions and "no E3" controls in D. Values are reported as means ± standard deviation ($n = 3$). Significance was determined by Student's *t*-test; *$P < 0.05$.

Source data are available online for this figure.

Similar to Ubc7 discharge assays, the rates of chain elongation reactions showed a clear dependence on the linchpin identity for both E3 ligases. As before, the E3 variant with an arginine linchpin yielded the highest reaction rates, followed by histidine, alanine, and glutamate; the latter again gave rates close to the "no E3" control. However, unlike in discharge assays, both wild-type E3s provide very similar stimulation of Ubc7-mediated elongation in each reaction step despite the difference in their linchpins (compare rates with black frames in each step). Thus, the difference in allosteric Ubc7 stimulation afforded by each E3 does not impact elongation reactions where the substrate is Ub at the end of a growing chain.

A comparison of the individual reaction steps revealed that the first reaction, i.e., mono-Ub to di-Ub, is the slowest with the next two reactions (di-Ub to tri-Ub and tri-Ub to tetra-Ub) showing generally comparable and higher rates (see Fig EV3B and C). The Ubc7 co-factor, Cue1, aligns the end of a growing Ub chain for modification by binding to the penultimate Ub moiety (von Delbrück *et al*, 2016). To parse out the contributions of Cue1 and the RING linchpin in Ubc7-mediated Ub chain formation, reactions were carried out with the Cue1(RGA) variant that carries a mutation in its Cue1 domain and thus renders it Ub binding-deficient and unable to align the growing Ub chain (Bagola *et al*, 2013). Similar to previous studies, the Cue1(RGA) mutant shows a large decrease in rates of reactions using chains of at least di-Ub as substrate (see di-Ub to tri-Ub and tri-Ub to tetra-Ub in Fig EV3B and C). In contrast, the Cue1 mutant has no effect on the reactions that generate di-Ub (see mono-Ub to di-Ub in Fig 3B; von Delbrück *et al*, 2016). Notably, in reactions carried out by each wild-type E3, the Cue1(RGA) mutation consistently reduces rates to the level of the mono-Ub to di-Ub reaction (highlighted as "minimal wt rates" in Fig 3B). The results demonstrate that, for elongation reactions beyond di-Ub, a defect caused by a RING linchpin mutation in the presence of wild-type Cue1 is smaller or equal to that of Cue1(RGA) in the presence of wild-type RINGs (in Fig 3B, compare rates of linchpin variants in di-Ub to tri-Ub reactions with rates of wild-type E3 in Cue1(RGA) background). The data also reveal that the first reaction of the elongation process is its kinetic bottleneck.

How are these behaviors reflected *in vivo*? We were unable to follow Ub chain formation on ERAD substrates *in vivo* in a

sufficiently quantitative manner, so we focused on its ultimate biological outcome, i.e., ERAD substrate degradation. Of course, substrate ubiquitylation (priming and chain formation) entails only two steps of many in this process (Fig 3C), but if one or both are rate-determining, changes in them are predicted to affect the observed rate of degradation. Pulse-chase and CHX decay assays were performed in yeast to follow ERAD-mediated protein degradation of vetted model substrates for each E3 (PrA*-3xHA and Hmg2-6xMyc for Hrd1; Deg1-eGFP$_2$ and FLAG-Sbh2 for Doa10—see Figs 3D, and EV4E and EV5E). Relative to strains expressing wild-type Cue1 and Hrd1 or Doa10, degradation of respective model substrates is markedly slower in strains lacking the respective E3 or Cue1. Given the well-documented integral role of the two E3s and of Cue1 in degradation of these substrates, the deletion strains report the stability of these substrates in the absence of ERAD-mediated degradation. In particular, the Hrd1 model substrate Hmg2-6xMyc (Fig EV4E) and the Doa10 model substrate Deg1-eGFP$_2$ (Fig 3D— right panel) are not completely stabilized in the Δcue1, Δhrd1, and Δdoa10 strains, respectively, indicating that these substrates can be degraded through other pathways.

Remarkably, in the presence of the relevant wild-type E3, strains that express the Cue1(RGA) mutant show similar model substrate degradation kinetics to those expressing wild-type Cue1 (Figs 3D and EV4E). Only in the case of the Doa10 model substrate FLAG-Sbh2 does the Cue1(RGA) mutation have a negative effect on protein degradation kinetics (Fig EV5E). This effect is, however, only modest when compared to those in the Δdoa10 and Δcue1 strains, and by the end of the assay, protein levels in the Cue1(RGA)-expressing strain were similar to those in the strain expressing the wild-type protein. The simplest explanation for this overall lack of an effect of the Cue1(RGA) mutant is that chain elongation is not rate-determining for ERAD protein degradation (see Fig 3C). Furthermore, we can conclude that the first reaction of chain elongation (mono-Ub to di-Ub) is not rate-determining for protein degradation *in vivo* at either E3 ligase because this step was the slowest *in vitro* and the Cue1(RGA) mutation slowed down all elongation steps to the rate of the first step. If the first step contributed to the experimentally observed rate of degradation, one would expect that additional reactions occurring at similar speeds (rather than much faster) would contribute to the overall rate. As this is not what we observed,

we conclude that none of the steps of chain elongation are rate-determining for substrate degradation as observed in our assays. This also suggests that the rates observed *in vitro* for the first step ("minimal wt rates") are a lower limit above which effective protein degradation at both E3s under the conditions assayed *in vivo* is guaranteed.

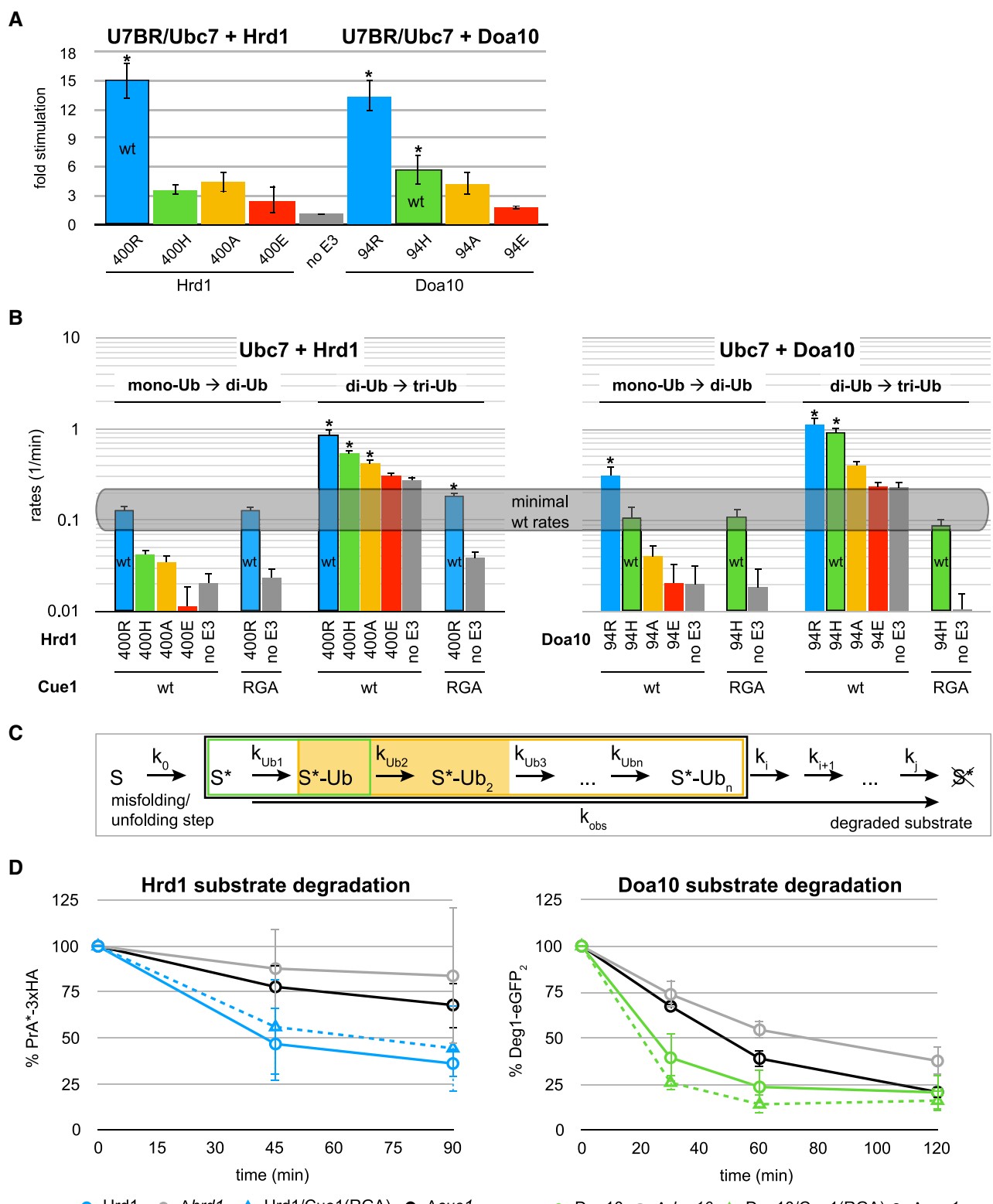

Figure 3.

◀

**Figure 3. Ubc7-mediated Ub chain elongation is not rate-determining for protein degradation.**

A   *In vitro* Ub nucleophile discharge assays for U7BR/Ubc7 with indicated Hrd1 (left) and Doa10 (right) variants. E2 stimulation is shown as the ratio of rates derived from RING-catalyzed reactions and the "no E3" control (see Fig EV2C and D). Values are reported as means ± standard deviation ($n = 3$). Significances for pairwise comparisons were determined by one-way ANOVA test; *$P < 0.05$. For clarity, only significances related to the "no E3" control are shown. In addition, reactions for Hrd1(400R) and Doa10(94R) are significantly faster than all other reactions and the reaction for Doa10(94H) is significantly faster than that for Doa10(94E). The wild-type E3s are identified by black frames for each set.

B   *In vitro* Ub chain formation assay by Ubc7 with indicated Hrd1 (left) and Doa10 (right) variants in the presence of indicated Cue1 variants. Rates for reactions of mono-Ub to di-Ub and di-Ub to tri-Ub with fluorescently labeled Ub are shown on a logarithmic scale. Values are reported as means ± standard deviation ($n = 3$). Significances for pairwise comparisons were determined by one-way ANOVA test; *$P < 0.05$. For clarity, only significances related to the "no E3" control of given reaction set are shown.

C   Steps of ERAD protein degradation with associated rate constants; S = ER protein; S* = misfolded protein, i.e., ERAD substrate; $k_{obs}$ = observed rate of protein degradation in assays; $k_{Ub1}$ = rate of the priming reaction; $k_{Ub2}$ = rate of the first elongation step, i.e., mono-Ub to di-Ub; $k_{Ub3} - k_{Ubn}$ = rates of subsequent elongation steps; $k_0 - k_j$ = rates for other steps of ERAD; the black box identifies steps of substrate poly-ubiquitylation with priming highlighted by the green box and chain elongation by the yellow box. The light yellow background highlights the mono-Ub to di-Ub reactions.

D   Protein degradation in indicated yeast strains monitored by pulse-chase experiments for the Hrd1 model substrate PrA*-3xHA (left) and by CHX decay assays for the Doa10 model substrate Deg1-eGFP$_2$ (right). Values for each time point are reported as means ± standard deviation ($n = 4$ for PrA*-3xHA and $n = 3$ for Deg1-eGFP$_2$).

Source data are available online for this figure.

## Linchpin-mediated Ubc7 stimulation during chain elongation is not crucial for ERAD substrate degradation

The revelation that Ub chain elongation is not rate-determining for substrate degradation implies two possibilities: (i) either the E3-dependent priming step is rate-determining or (ii) a process other than substrate poly-ubiquitylation and independent of an E3 is (see Fig 3C). To distinguish the two possibilities, linchpin mutants of Hrd1 and Doa10 were introduced into yeast. As Ubc7 stimulation depends on the linchpin (see Fig 3A), linchpin variants might allow us to uncouple the recruitment function of each E3 ligase from their E2 stimulation function. In pulse-chase experiments tracking Hrd1 model substrate degradation, we observed a dependence of substrate degradation on the identity of the Hrd1 linchpin (Figs 4A —left panel and EV4A) with similar trends to those *in vitro*: Substrate degradation occurs most rapidly in the strain expressing wild-type Hrd1 with its arginine linchpin, followed by histidine and alanine linchpin mutants. Expression of Hrd1 with a glutamate linchpin results in degradation impaired to a degree comparable to a strain that lacks Hrd1 altogether.

The fact that substrate degradation rates are detectably altered by introduction of Hrd1 linchpin mutations suggests that there is a Hrd1-dependent step that is rate-determining. In *in vitro* chain elongation assays, Hrd1 linchpin mutations slow the first step below the lower limit that would guarantee unimpeded protein degradation (see Fig 3B—left panel). Therefore, we can unambiguously conclude from experiments with the linchpin variants that linchpin-mediated stimulation of Ubc7 by Hrd1 is involved in a rate-determining step of protein degradation. But we cannot say whether this is the priming step or the first step of chain elongation (or both) based on the data for Hrd1 alone. As Doa10 uses different E2s for the two steps in question, we turned to Doa10 for clarification.

The effects of Doa10 linchpin mutations on protein degradation were assessed by introducing individual Doa10 variants in the genomic *DOA10* locus and analysis of protein degradation by CHX decay assays. In stark contrast to the Hrd1 case, degradation rates of Doa10 substrates are not altered for Doa10 linchpin variants (Fig 4A —right panel). Similar to the case for the Cue1 mutant, the model substrate FLAG-Sbh2 appears to be an exception, as its degradation rates are decreased in strains carrying either Doa10(94R) or (94A). However, degradation was still substantially faster than in the strain

deleted for Doa10 (Fig EV5A). The overall lack of effects in observed degradation rates for Doa10 linchpin mutants implies that these mutations do not slow chain elongation to an extent that adversely affects protein degradation. But the Doa10 linchpin mutations substantially impede the rate of the slowest mono-Ub to di-Ub reaction in *in vitro* chain elongation reactions to well below the threshold "minimal wt rates" (see Fig 3B—right panel). This means that our previously specified lowest rate estimate for chain elongation can be seriously undercut without resulting in an observed change in the rate of protein degradation observed in our assays. Importantly, the rates observed for Hrd1 linchpin mutants are similar to those of Doa10 linchpin mutants for the first step of elongation (see Fig 3B). Altogether, the results allow us to infer that Hrd1 linchpin mutants do not impede degradation rates through effects on chain elongation. We therefore propose that priming is the E3-dependent rate-determining step in protein degradation in the presence of Hrd1 linchpin variants.

## Linchpin-mediated E2 stimulation is crucial for priming at Hrd1, but not at Doa10

To test our hypothesis further, Hrd1 linchpin mutants were tested for their ability to prime in the *in vitro* substrate ubiquitylation assay (Fig 4B). In contrast to the robust RNase modification observed for Ubc7 with wild-type S-Hrd1(400R), only faint or undetectable RNase-Ub bands appeared with Hrd1 linchpin mutants. The concurrence of Hrd1 linchpin effects in both *in vivo* substrate degradation assays and *in vitro* ubiquitylation assays is strong evidence that priming of substrates and not Ub chain elongation is the slower and thus rate-determining step for Hrd1-dependent ERAD substrate degradation. The results also reveal that linchpin-mediated E2 stimulation by Hrd1 is crucial for priming, but is dispensable for elongation *in vivo*. This is consistent with the ability of Doa10 with its suboptimal linchpin to function effectively with Ubc7 in the chain elongation stage of its substrate ubiquitylation process.

The apparent linchpin independence of Doa10 substrate degradation rates might suggest that its priming step is also linchpin-independent. To test this hypothesis, we performed *in vitro* Ub discharge assays of Doa10 linchpin mutants with Doa10's preferred priming E2, Ubc6. No clear correlation between linchpin identity and ability to stimulate Ubc6 was observed (Fig 4C). Indeed, while

the non-native arginine linchpin is most effective at stimulating Ubc6 discharge, the wild-type histidine linchpin is least effective and alanine and glutamate linchpins appear slightly more effective than the wild-type. Moreover, Ubc6 stimulation does not show any correlation with linchpin identity, when interacting with Hrd1 (Fig EV2F and G), thus strongly supporting linchpin insensitivity as

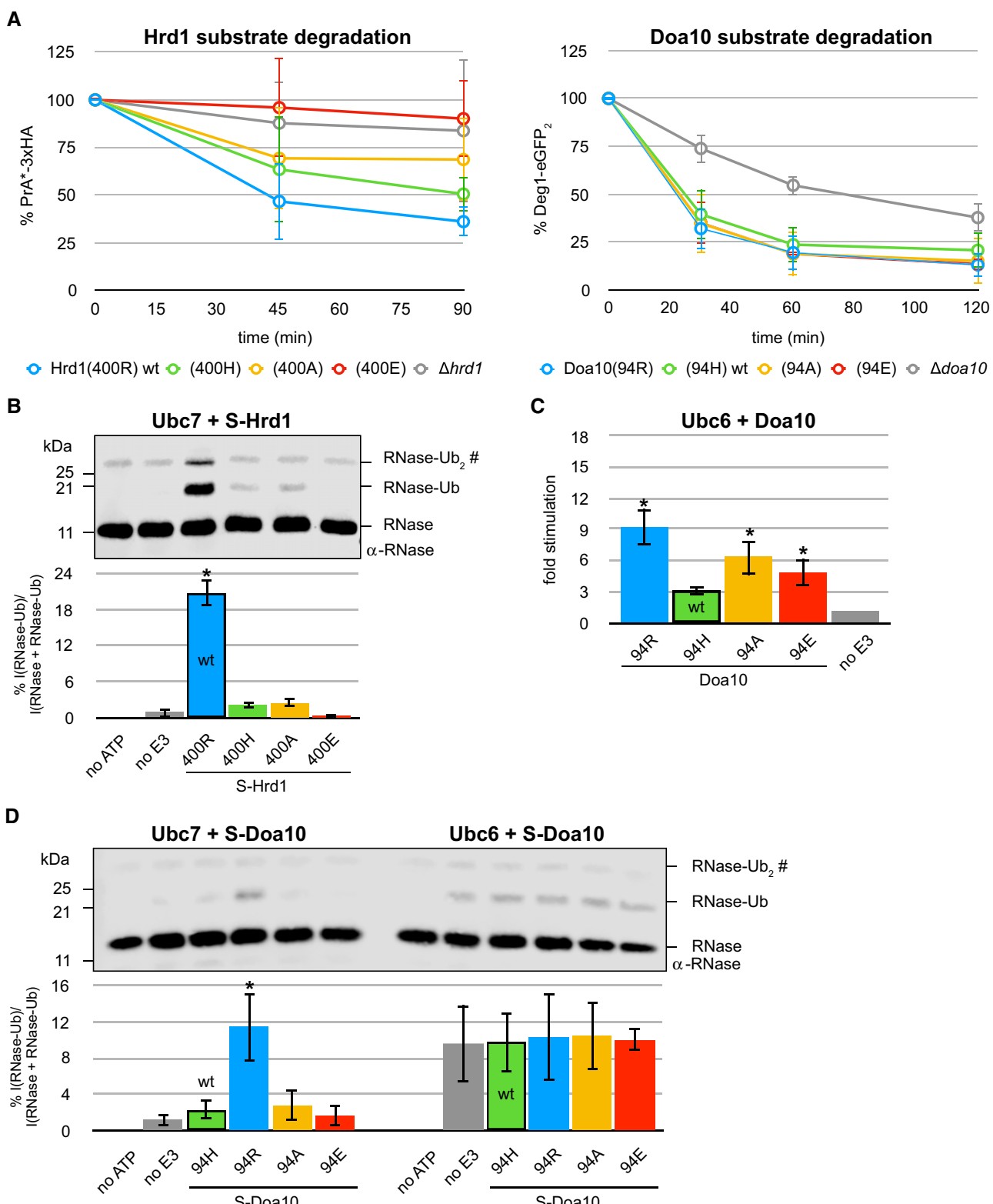

**Figure 4.**

◄

**Figure 4.  Linchpin-mediated E2 stimulation is crucial for priming at Hrd1, but not at Doa10.**

A  Protein degradation in indicated yeast strains monitored by pulse-chase experiments for the Hrd1 model substrate PrA*-3xHA (left) and by CHX decay assays for the Doa10 model substrate Deg1-eGFP$_2$ (right). Values for each time point are reported as means ± standard deviation ($n$ = 4 for PrA*-3xHA and $n$ = 3 for Deg1-eGFP$_2$). Data for control strains (Hrd1(400R) wt, *Δhrd1*, Doa10(94H) wt, and *Δdoa10*) are the same as in Fig 3D. A side-by-side comparison of protein degradation in all tested strains is shown in Appendix Figs S1 and S2.

B  *In vitro* substrate ubiquitylation assay for Hrd1 variants with Ubc7. Ubc7 reactions contained equimolar amounts of Cue1 and were performed with Ub(K48R). Top: Representative immunoblot using a poly-clonal α-RNase A antibody; "no E3" reaction does not contain any S-Hrd1, "no ATP" reaction does not contain ATP, but wild-type S-Hrd1(400R). The RNase-Ub$_2$ band co-migrates with a non-specific band (#) common to all samples. Bottom: Quantification of RNase-Ub signals is shown. Values are reported as means ± standard deviation ($n$ = 3). Significances for pairwise comparisons were determined by one-way ANOVA test; *$P$ < 0.05. For clarity, only significances related to the "no E3" control are shown. In addition, the reaction containing S-Hrd1(400R) shows significantly more ubiquitylation than those with other E3 variants.

C  *In vitro* Ub nucleophile discharge assays for Ubc6 with indicated Doa10 variants. E2 stimulation is shown as the ratio of rates derived from RING-catalyzed reactions and the "no E3" control (see Fig EV2E). Values are reported as means ± standard deviation ($n$ = 3). Significances for pairwise comparisons were determined by one-way ANOVA test; *$P$ < 0.05. For clarity, only significances related to the "no E3" control are shown. In addition, the reaction for Doa10(94R) is significantly faster than that for Doa10(94H) and (94E). The wild-type E3 is identified by a black frame. The scale of the $y$-axis was set identical to that of plots from other discharge assays (see Figs 2E and 3A) for easier comparison.

D  *In vitro* substrate ubiquitylation assay for Doa10 variants with Ubc7 (left) and Ubc6 (right). Ubc7 reactions contained equimolar amounts of Cue1 and were performed with Ub(K48R). Top: Representative immunoblot using a poly-clonal α-RNase A antibody; "no E3" reaction does not contain any S-Doa10, "no ATP" reaction does not contain ATP, but wild-type S-Doa10(94H). The RNase-Ub$_2$ band co-migrates with a non-specific band (#) common to all samples. Bottom: Quantification of RNase-Ub signals is shown. Values are reported as means ± standard deviation ($n$ = 3). Significances for pairwise comparisons were determined by one-way ANOVA test; *$P$ < 0.05. For clarity, only significances related to the "no E3" control of a given E2 are shown. In addition, the reaction containing Ubc7 and S-Hrd1(400R) shows significantly more ubiquitylation than those with Ubc7 and other E3 variants.

Source data are available online for this figure.

an intrinsic feature of Ubc6. While this lack of linchpin dependency appears congruent with the observation that degradation of Doa10-dependent substrates is unaffected by Doa10 linchpin mutations (see Fig 4A—right panel), we cannot rigorously conclude that the priming step of Doa10-catalyzed substrate ubiquitylation is rate-determining. The formal possibility that a step that does not involve Doa10 is rate-determining for its substrate degradation cannot be ruled out by the combination of *in vivo* degradation and *in vitro* discharge assays for Doa10. Based on the fact that poly-ubiquitylation and, specifically, priming by Ubc7 are rate-determining for Hrd1-dependent degradation, it is likely that the same is true for Doa10-dependent priming by Ubc6. Importantly, however, for both Doa10 and Hrd1 substrate degradation, the Ubc7-dependent chain elongation is not rate-determining.

Data presented thus far provide a rationale for why Ubc7 is not the functional priming partner for Doa10. The question remains how Doa10 is able to function with Ubc6. To assess how Ub transfer to a substrate (as opposed to nucleophile) is affected by the Doa10 linchpin, we performed *in vitro* assays with both E2s and S-Doa10 linchpin variants (Fig 4D). When paired with Ubc7, only the hyperactive Doa10(94R) variant produces product at a level significantly above the "no E3" reaction. Thus, in this proxy for a priming reaction, wild-type Doa10 is ineffective at stimulating Ubc7 to transfer Ub to a substrate. In Ubc6 reactions, product bands of comparable intensity were detected in all reactions containing Ubc6, including the reaction with no E3 present. Hence, the assay does not provide clarity regarding the linchpin contribution, or lack thereof, for priming reactions carried out by Ubc6. Nevertheless, the assay does recapitulate the strong dependence of Ubc7 on an optimal linchpin. Given the robust RNase modification observed for the S-Doa10(94R) variant with Ubc7, it appears that the suboptimal wild-type histidine linchpin of Doa10 actively disfavors use of Ubc7 for priming, leaving the way clear for pairing with Ubc6.

In the substrate ubiquitylation assays (see Fig 4D), reactions carried out in the absence of E3 reveal a difference between the ability of Ubc6 and Ubc7 to transfer Ub without E3 stimulation. While

no to little product was generated by Ubc7, product generated by Ubc6 was clearly detected. Moreover, similar amounts of product were generated in reactions that included Doa10. These observations suggest a scenario where priming can be carried out by Ubc6 without an absolute requirement that its Ub transfer activity be stimulated by an E3. In contrast, regulation of the priming function of Ubc7 appears to depend critically on its linchpin-dependent stimulation by an E3 (as shown for Hrd1—see Fig 2).

### The Ubc6~Ub conjugate populates closed conformations more than other E2~Ub conjugates

The functional preference of Doa10 for Ubc6 over Ubc7 during priming could be determined solely by Ubc7's dependence on an optimal linchpin. But it raises the question of how Ubc6 circumvents the disadvantage of Doa10's suboptimal histidine linchpin. Several observations point to a high basal (i.e., non-E3-stimulated) activity for Ubc6. First, Ubc6 has higher intrinsic ubiquitylation activity on the proxy substrate RNase (compare "no E3" lanes between Ubc6 and Ubc7 in Figs 2B and 4D). Second, the absolute rates of Ub discharge onto ethanolamine in the absence of an E3 are higher for Ubc6 compared to U7BR/Ubc7 (compare "no E3" controls in Figs EV2C and D with EV2E and F). This observation must be interpreted with care due to the bi-functional nature of ethanolamine and the different reactivity profiles of the two E2s. Third, the rates of hydrolysis (i.e., Ub discharge via H$_2$O) of Ubc6 and U7BR/Ubc7 conjugates are vastly different (Fig 5A). And although reactivity profile differences may play a role in this case too, stability toward hydrolysis has been used as a general measure of E2 reactivity (Pruneda *et al*, 2011; Plechanovová *et al*, 2012). In sum, the observations indicate that Ubc6~Ub has higher intrinsic reactivity than Ubc7~Ub.

To understand the source of Ubc6's higher intrinsic activity, disulfide-linked Ub conjugates of both E2s were characterized by NMR. We focused on the spectral properties of the common subunit, i.e., Ub, in the two species, collecting NMR spectra on samples in which only the Ub subunit was isotopically labeled

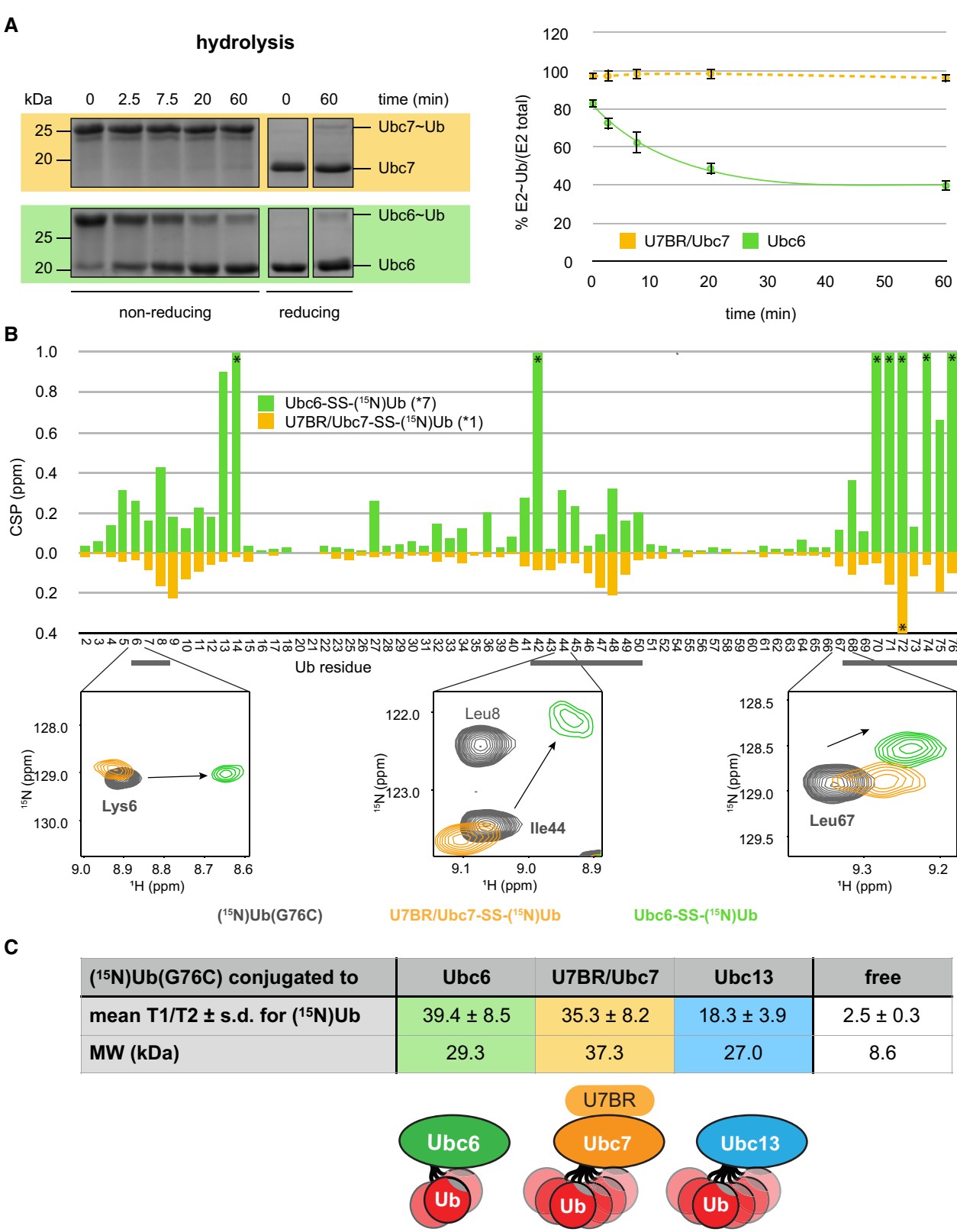

**Figure 5.**

◀

**Figure 5. The Ubc6~Ub conjugate populates closed conformations more than other E2~Ub conjugates.**

A E2~Ub hydrolysis assays for U7BR/Ubc7 and Ubc6. E3-independent Ub discharge assays for both E2s in PBS (absence of nucleophile = hydrolysis) were performed at 32°C. Left: Representative Coomassie gels. Right: Plots of E2~Ub hydrolysis (dots) as a function of time with first-order reaction models fitted to the discharge data (lines) are shown. Values for each time point are reported as mean $\pm$ standard deviation ($n$ = 3).

B HSQC-TROSY experiments comparing free and E2-conjugated ($^{15}$N)Ub. Top: CSPs of ($^{15}$N)Ub(G76C) conjugated to indicated E2s via disulfide bond (775 µM each) compared to free ($^{15}$N)Ub(G76C) are shown; Ub residues known to be involved in the closed conformation interface are underlined in gray; *number of resonances with perturbations too large to be assigned confidently (shown as off-scale in diagram). Bottom: Spectra of resonances for selected residues in Ub's hydrophobic patch for free Ub(G76C) (gray) and Ub conjugated to Ubc6 (green) and U7BR/Ubc7 (yellow).

C Summary of T1 and T2 relaxation analysis for ($^{15}$N)Ub(G76C) in various E2 conjugates. The table shows the mean T1/T2 ratios $\pm$ standard deviation over all Ub peaks in a given conjugate (or for the free protein) as well as their molecular weights. Bottom: Cartoons depict open/closed conformation equilibria for different conjugates.

Source data are available online for this figure.

(E2-SS-($^{15}$N)Ub). The spectra of Ub in the context of each conjugate were compared to the spectrum of free Ub, and differences are reported as chemical shift perturbations (CSPs). It has been established for other E2~Ub species that the more Ub inhabits "open" conformations, the closer its spectrum will resemble that of free Ub (i.e., small CSPs) (Hamilton *et al*, 2001; Pruneda *et al*, 2011, 2012; Wickliffe *et al*, 2011; Choi *et al*, 2015; Dove *et al*, 2016). As the populations or lifetimes of closed conformations increase, observed CSPs on Ub increase due to the altered chemical environment at the E2/Ub interface. Ub CSPs that center around Leu8, Ile44, Val70, and the C-terminal tail are observed in both the Ubc6 and Ubc7 conjugates (Fig 5B). Leu8, Ile44, and Val70 define the Ub hydrophobic patch that interacts with an E2 in reactive closed conformations (see Fig 1B), and the C-terminal residues link Ub to the E2 active site. The striking difference between the two conjugates is the very large magnitude of Ub CSPs in the Ubc6 conjugate (Fig 5B and Appendix Figs S3 and S4), which are easily among the largest published to date. Seven peaks are either lost due to exchange broadening or shift to an extent that they cannot be assigned with confidence in the spectrum of Ubc6-SS-($^{15}$N)Ub as compared with only one such perturbation in the U7BR/Ubc7-SS-($^{15}$N)Ub spectrum. The data reveal that Ub conjugated to Ubc6 experiences environments that are more dissimilar to free Ub than Ub in conjugates of Ubc7 or other E2s studied to date (see spectral insets in Fig 5B and Appendix Fig S3 and examples in Choi *et al*, 2015; Dove *et al*, 2016; Pruneda *et al*, 2011). This is strong evidence that a greater population of Ubc6~Ub resides in closed conformations compared to other E2~Ub conjugates and particularly Ubc7~Ub in the absence of an E3 ligase.

$^{15}$N relaxation parameters were measured as an independent assessment of Ub in different conjugates. Ratios of $^{15}$N relaxation times, T1/T2 (Fig 5C), provide a metric for how flexible Ub is in each conjugate (Pruneda *et al*, 2011). For globular proteins, T1/T2 ratios scale with molecular weight, as they report on molecular tumbling time. Despite Ubc6 having a lower molecular weight than U7BR/Ubc7, their conjugated Ubs' average T1/T2 values are similar (39.5 $\pm$ 8.2 vs. 36.6 $\pm$ 9.8, respectively; Fig 5C). For context, T1/T2 ratios were also determined for Ub attached to human Ubc13 which is much closer in molecular weight to Ubc6 than U7BR/Ubc7. Here, the much smaller average ratio (18.2 $\pm$ 3.9) is congruent with a highly flexible Ub that predominantly inhabits a myriad of open conformations, as established previously by other techniques (Pruneda *et al*, 2011). Taken together, these values are strong evidence that Ub attached to Ubc6 spends more time in conformations in which the conjugate tumbles as a single unit rather than

two tethered but independent molecular entities. In combination with the large CSPs observed at Ub's hydrophobic patch, the relaxation data strongly support the notion that the Ubc6~Ub conjugate is more biased toward closed conformations than those of other E2s.

A bias toward closed conformations is important for two reasons: first, it is consistent with the greater intrinsic Ub transfer activity we observed for Ubc6~Ub, as closed conformations are generally associated with increased reactivity. In line with this is the observation that E3s only modestly stimulate Ubc6 from its already high basal activity as compared to Ubc7 (see Fig EV2G). Second, the closed conformation bias of Ubc6~Ub helps explain why the conjugate is less dependent on allosteric stimulation via a RING linchpin than Ubc7~Ub. The results thus provide an explanation for why Ubc6 does not strictly rely on a linchpin and, consequently, how Doa10 with its suboptimal linchpin is able to pair functionally with Ubc6 for priming.

**Versatility in ERAD priming E2 usage**

Despite their established E2 preferences, we have shown that both ERAD E3s can work with their non-preferred priming E2, at least *in vitro*. This raises the question as to whether there is some crossover of priming E2 utilization *in vivo*. To address this point, we examined ERAD substrate turnover in cells that lack either Ubc6 or Ubc7. Consistent with the essential role of Ubc7 in the Ub chain elongation step, degradation of most substrates monitored is abolished in cells lacking Ubc7 to levels of cells deleted for the E3 ligases (Figs 6A–C and EV4C—compare solid and dotted curves). Hmg2-6xMyc (Fig EV4C) and Deg1-eGFP$_2$ (Fig 6B) degradation still occurs in $\Delta ubc7$ strains, as it does in $\Delta hrd1$ and $\Delta doa10$ strains, respectively (see also Fig 3D), suggesting an alternative pathway of protein quality control for these substrates. As predicted, deletion of *UBC6* has no effect on degradation of Hrd1 model substrates (Figs 6D and EV4D—compare solid and dashed blue curves). However, in strains expressing a suboptimal, Doa10-like Hrd1 linchpin mutant (400H), deletion of *UBC6* has a slight adverse impact on the degradation of the Hrd1 model substrate PrA*-3xHA (Fig 6D—compare solid and dashed green curves). In this Hrd1 hypomorphic setting, the absence of Ubc6 impairs degradation to the level of strains lacking Hrd1 altogether. A similar, though less pronounced trend was detected for Hmg2-6xMyc degradation (Fig EV4D). These observations imply that Hrd1 can use Ubc6 for priming *in vivo* at least to a small degree and/or under extreme conditions.

To pursue the potential of Ubc6 priming at Hrd1 further, we turned to published evidence that genetically linked Ubc6 to Hrd1 for degradation of its model substrate, CPY* (Hiller *et al*, 1996). To

confirm involvement of Ubc6 unambiguously, we leveraged the ability of this E2 to transfer Ub to hydroxyl-containing side chains (Weber *et al*, 2016) by using a lysine-less version of CPY* ("CPY*-K0-HA") (Baldridge & Rapoport, 2016). We confirmed that CPY*-K0-HA is not a substrate for Doa10, as its degradation rate is not affected by deletion of *DOA10* (Fig 6E—compare black vs. purple curves). Nevertheless, deletion of *UBC6* is as detrimental as deletion of *HRD1* or introduction of a catalytically dead version of Ubc7 (C89S) (Fig 6E—compare green vs. blue and yellow curves). Furthermore, simultaneous deletion of *HRD1* and *UBC6* does not result in an additional effect (Fig 6E—compare green vs. gray curves), suggesting Hrd1 and Ubc6 function in the same pathway for this substrate. Thus, the model Hrd1 substrate can be primed by Ubc6 in its hydroxyl-reactive mode.

In a similar vein, we asked whether Doa10 substrates can be primed by Ubc7, despite the suboptimal linchpin. As predicted by the functional pairing of Ubc6 and Doa10, we observed a negative impact on the degradation of Doa10 model substrates with deletion of *UBC6* (Fig 6B and F—compare solid vs. dashed green curves). Degradation of the Doa10 model substrate Deg1-eGFP$_2$ is impaired to similar levels in cells deleted for *DOA10* or *UBC6* (Fig 6B), illustrating that processing of this Doa10 substrate is strictly dependent on priming by Ubc6. However, cells that lack Ubc6 are still able to turnover FLAG-Sbh2, the other Doa10 model substrate studied here, albeit at a slower rate than when the favored priming E2 is present (Fig 6F—compare solid and dashed green curves). This suggests that Ubc7 can carry out priming at Doa10 E3 ligase. However, when we repeated the experiment in a yeast strain that lacks Ubc6 and expresses the Ubc7-activating Doa10(94R) variant, we did not observe improved FLAG-Sbh2 turnover (Fig 6F—compare dashed green and blue curves). Curiously, in a strain that expresses Ubc6 and the hyperactive Doa10(94R) variant, FLAG-Sbh2 degradation is even slightly impaired (Fig 6F—compare solid green and blue curves) suggesting that *in vivo* the situation is more complex and Doa10 cannot switch seamlessly between priming E2s depending on its linchpin. Nevertheless, the data illustrate that "suboptimal" E2/E3 pairings are possible and can carry out priming reactions in cells.

## Discussion

In protein poly-ubiquitylation, a distinction between the priming step of mono-ubiquitylation and subsequent elongation steps of poly-Ub chain building is conceptually acknowledged. The notion that these distinct processes would be carried out by different E2s was slow to be embraced, but is now well established for multiple systems (Rodrigo-Brenni & Morgan, 2007; Parker & Ulrich, 2009; Wu *et al*, 2010a; Weber *et al*, 2016). For many systems, however, the identity of the relevant priming E2(s) remains unexplored and even in cases where the complete set of E2s that can function with a given E3 is known, which pairing(s) are functionally relevant in a given cellular situation is not. For example, the heterodimeric RING E3, BRCA1/BARD1, was shown to work with five priming E2s and three elongating E2s (Christensen *et al*, 2007), but the pairings relevant for cellular function remain to be determined. Lack of specific information regarding which E2 is used by which E3 for which substrate and for which functions (priming and/or elongation) have impeded efforts to understand functional E2/E3 interplay in greater detail.

As it has been so thoroughly studied, the ERAD poly-ubiquitylation system provided a powerful way to address the question of how an E3 pairs with its preferred E2(s) for priming and elongation. Several features of the ERAD system were key. First, its two E3 ligases have non-overlapping substrate specificities: Hrd1 mainly targets proteins with lesions in their luminal or membrane domains, while Doa10 predominantly degrades proteins with cytosolic lesions (Ruggiano *et al*, 2014). This allowed us to monitor the effects of perturbations on substrates of one or the other E3. Second, the two E3s use the same chain-elongating E2, Ubc7, but differ in their choice of priming E2. Third, both E3 ligases retain the ability to pair with the non-preferred E2 for priming (i.e., Hrd1 with Ubc6 and Doa10 with Ubc7). While this property can confound interpretation of *in vivo* studies, it provided important controls in our investigation.

Through a combination of *in vitro* and *in vivo* experiments, we have determined the mechanism how each of the ERAD RING E3 ligases utilizes their preferred E2 enzyme during priming. Hrd1 selects Ubc7 over Ubc6 through higher binding affinity: a mechanism we dub "affinity-driven pairing". The greater than one order-of-magnitude difference in affinity guarantees that a given Hrd1 RING will engage Ubc7~Ub more frequently and with a longer lifetime than a similarly disposed Ubc6~Ub, particularly because both E2s appear to be expressed at roughly similar levels [according to PaxDb—(Wang *et al*, 2015) and (Clague *et al*, 2015)]. Furthermore, Hrd1 binding stimulates the Ub transfer activity of Ubc7 better than it does Ubc6 (see Fig 2E). Together, the affinity and subsequent stimulation provide a strong selective preference of Hrd1 for Ubc7 (Fig 7—top). This is in line with the previously reported differential stimulation of Ubc7 by Hrd1 and Doa10 (Cohen *et al*, 2015). Thus, Hrd1 seems to follow the classical E3 dualism of using its E2 recruitment *and* stimulation function to control the priming reaction.

In contrast, Doa10 collaborates with different E2s for priming and elongation and does not seem to differentiate between the two based on affinity. Our study reveals that Doa10's use of Ubc6 during priming is established through a mechanism we call "rate-driven pairing" (Fig 7—bottom). It relies on two features: (i) Doa10's suboptimal histidine linchpin residue, with which the E3 largely abdicates its E2 stimulation function and thus disfavors collaboration with Ubc7 and (ii) Ubc6's relative linchpin insensitivity, which allows its utilization by Doa10 despite its suboptimal linchpin. The linchpin insensitivity stems from Ubc6's high basal activity due to the adoption of reactive closed conformations of its conjugate; hence, the E2 undergoes only moderate stimulation by either E3 *in vitro*. High E3-independent Ubc6 activity might pose a danger to a cell through ubiquitylation events that are outside of E3 control (as exemplified *in vitro* for the E3-independent ubiquitylation of RNase by Ubc6—see Fig 2B). We speculate that the reported constitutive auto-ubiquitylation of Ubc6 is a symptom of this as well as a means to keep this danger in check, as the auto-modification is Doa10-independent (Weber *et al*, 2016) and leads to rapid Ubc6 degradation in the absence of an ERAD substrate (Walter *et al*, 2001). Such a regulatory mechanism would fit with our finding that Ubc6 is a highly active E2 and underscores the importance of the substrate recruitment function of Doa10 as opposed to its stimulation function as means of spatiotemporal control of the priming reaction on an E3-bound substrate.

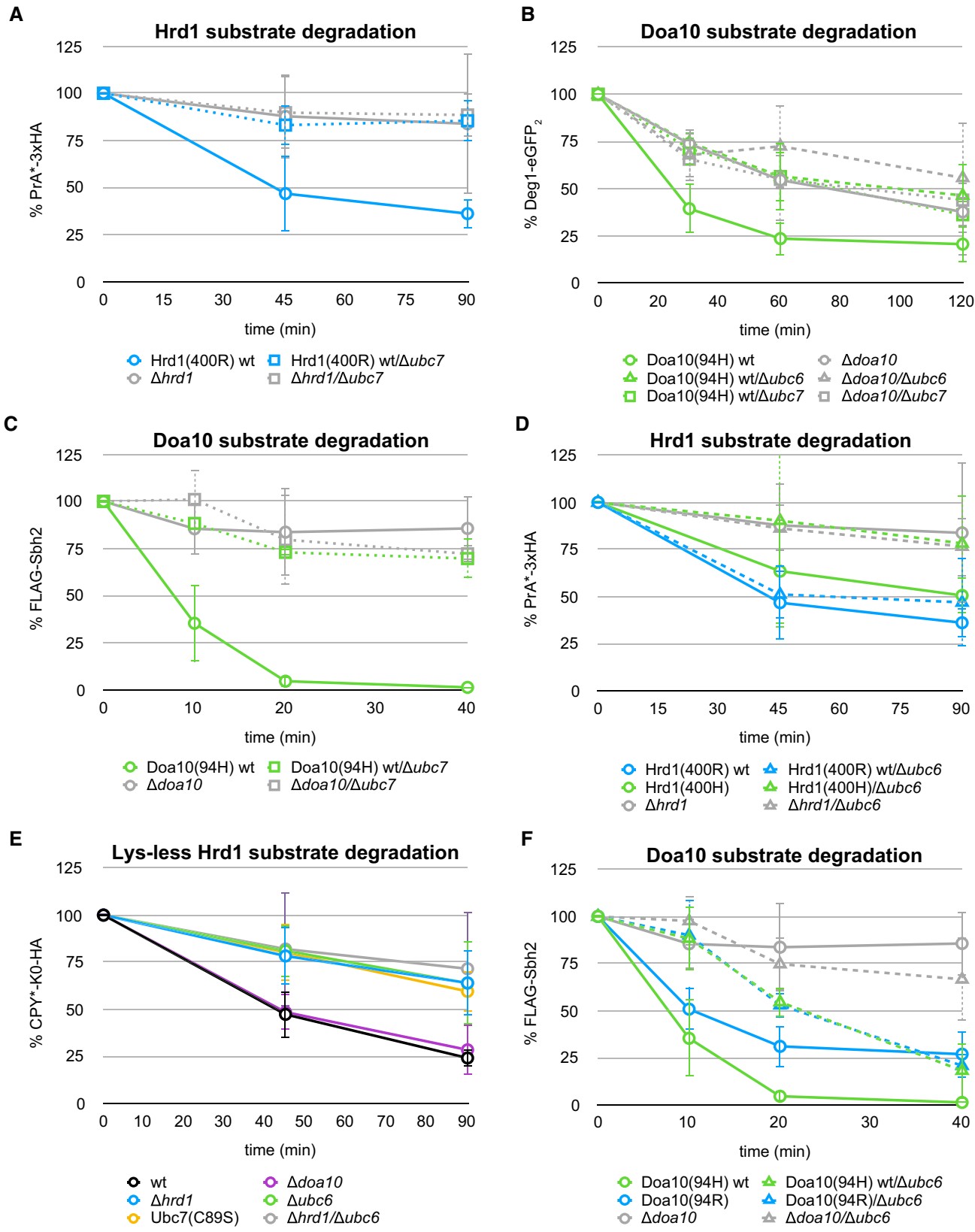

**Figure 6.**

**Figure 6.   Versatility in ERAD priming E2 usage.**

A–F   Protein degradation in indicated yeast strains monitored by pulse-chase experiments for the Hrd1 model substrates PrA*-3xHA (A and D) and lysine-less CPY*-K0-
HA (E) and by CHX decay assays for the Doa10 model substrates Deg1-eGFP$_2$ (B) and FLAG-Sbh2 (C and F). Values for each time point are reported as
means $\pm$ standard deviation ($n$ = 4 for PrA*-3xHA and $n$ = 3 for Deg1-eGFP$_2$ and FLAG-Sbh2, respectively). For A, data for control strains (Hrd1(400R) wt, Hrd1
(400H), and $\Delta$hrd1) are the same as in Figs 3D and 4A. For D, data for control strains (Doa10(94H) wt and $\Delta$doa10) are the same as in Figs 3D and 4A. A side-by-
side comparison of protein degradation in all tested strains is shown in Figs EV4 and EV5 and Appendix Figs S1 and S2.

Source data are available online for this figure.

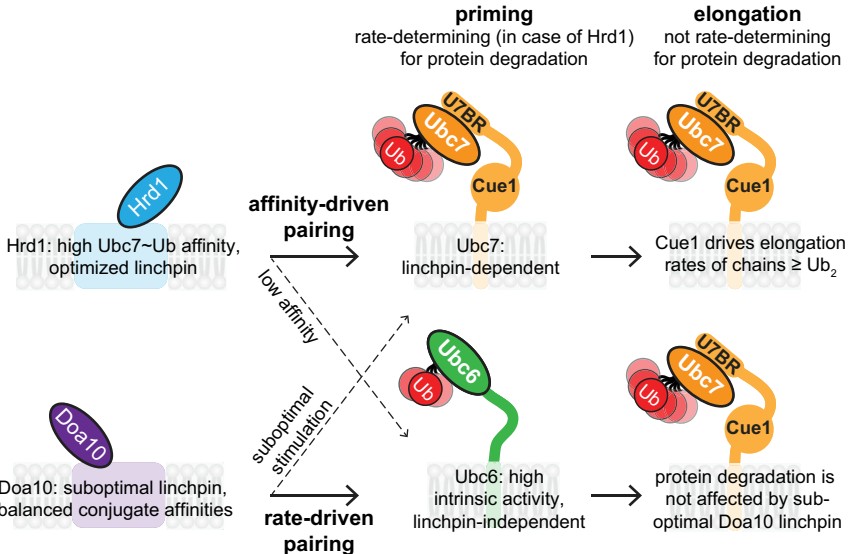

**Figure 7.   Model for E2 utilization by Hrd1 and Doa10.**

Hrd1 uses Ubc7 for both priming and elongation and selects Ubc7 over Ubc6 for priming through higher binding affinity and optimal stimulation through its arginine
linchpin ("affinity-driven pairing"). Doa10 uses Ubc6 for priming. Although it binds the conjugates of the two E2s with similar affinity, its suboptimal histidine linchpin is
ineffective at stimulating Ubc7. In contrast, Doa10 pairing with Ubc6 is highly active due to the high basal activity of Ubc6~Ub that does not require substantial stimulation
by a linchpin. We call this mode of E2 selection "rate-driven pairing". E3-mediated E2 stimulation is dispensable during elongation, allowing Doa10 to pair productively with
Ubc7. For later elongation steps, Cue1 plays an important role in driving reaction rates although Ub chain elongation is generally not rate-determining for the kinetics of
protein degradation. Finally, both RINGs retain the ability to use the non-preferred E2 for priming (dashed arrows), possibly providing a "backup mechanism" for priming.

Integral to the rate-driven pairing mechanism for Doa10 and
Ubc6 is our finding that the suboptimal linchpin of the RING does
not handicap its ability to use Ubc7 for subsequent Ub chain elon-
gation reactions. The Doa10/Ubc7 pair achieves rates of chain
elongation on par with those of the high-affinity Hrd1/Ubc7 pair
*in vitro* (see Fig 3B). Moreover, we could show that Ubc7-mediated
chain elongation is comparably fast and thus is not rate-deter-
mining to the overall process of ERAD substrate degradation.
Linchpin-mediated Ubc7 stimulation by the RINGs and chain align-
ment by Cue1 play a role in boosting elongation rates. Yet even
their disruption (through linchpin mutation or the Cue1(RGA)
mutant, respectively) does not slow elongation to a level where it
might become rate-determining for protein degradation. Our work
thus provides an explanation for the previously reported, yet unex-
plained discrepancy between the defect that Cue1(RGA) causes in
the *in vitro* situation and its apparent lack of an effect on protein
degradation *in vivo* (Bagola *et al*, 2013; von Delbrück *et al*, 2016).
Overall, our results demonstrate that, while the components
involved in the elongation reaction each contribute to optimizing
the process, chain elongation itself is not rate-determining to the
ultimate biological outcome of protein degradation.

Our investigation adds the ERAD system to those poly-ubiquity-
lation systems that have been studied to a level of detail that
provides kinetic insights on the different steps. Most prominent
among these systems are the mitotic master-regulator RING E3
complex APC/C (Rodrigo-Brenni & Morgan, 2007; Jin *et al*, 2008;
Summers *et al*, 2008; Garnett *et al*, 2009; Williamson *et al*, 2009;
Wu *et al*, 2010b; Brown *et al*, 2014, 2015; Kelly *et al*, 2014; Chang
*et al*, 2015; Lu *et al*, 2015) and the SCF RING E3 ligases complexes
(Kleiger *et al*, 2009; Wu *et al*, 2010a; Emberley *et al*, 2012; Pierce
*et al*, 2013). For the latter example, millisecond kinetic measure-
ments revealed that most enzyme–substrate encounters with the
SCF ligase are non-productive due to the low rate of the first Ub
transfer, while rates of sequential Ub additions were markedly faster
than substrate dissociation (Pierce *et al*, 2009). Qualitatively, this
kinetic dichotomy is reminiscent of the one we found in the ERAD
system.

As seen previously, we find that Hrd1's selection of Ubc7 and
Doa10's pairing with Ubc6 during priming is essential for robust
substrate degradation *in vivo* (Figs 6 and EV4F; Weber *et al*, 2016).
We also found that the ERAD E3s retain the ability to collaborate
with their non-preferred E2 for priming. Our work provides a

mechanistic framework for how preferred E2/E3 pairings are established but does not answer the question why these preferences are functionally important for ERAD substrate degradation. Using the linchpin dependencies uncovered, we could alter the E2/E3 pairings *in vitro*. For example, switching the Doa10 linchpin to arginine creates an E3 that can efficiently ubiquitylate the substrate proxy RNase via Ubc7 (see Fig 4D). Likewise, swapping Hrd1's linchpin to histidine disables its ability to stimulate Ubc7 (see Fig 3A), while its Ubc6 stimulation remains virtually unchanged (see Fig EV2G). But the same mutations do not translate into a seamless switch of priming E2 preference when viewed through the lens of protein degradation *in vivo*: While the Hrd1(400H) variant renders Hrd1 substrate degradation somewhat Ubc6-dependent (see Figs 6D and EV4D), it was still less efficient than with wild-type Hrd1(400R), which entirely depends on Ubc7. Similarly, degradation of Doa10 model substrates did not become Ubc6-independent in strains that carry the Ubc7-amiable Doa10(94R) mutant (see Fig EV5D and Appendix Fig S2D). Clearly, there is more to priming E2 preference than the effectiveness of an E3's linchpin, e.g., ternary interactions between the E3 ligase, its preferred E2 conjugate, and a specific substrate. Thus, correct E2/E3 pairing for priming fundamentally matters for efficient substrate degradation. As to the ability of both E3s to potentially use their non-preferred E2 for priming, we speculate that it might work as a failsafe mechanism under extreme conditions, e.g., Hrd1's use of Ubc6 for substrates with few or inaccessible lysines (see Fig 6E).

To summarize, we have provided a mechanistic and conceptual framework for how ERAD RING E3s exploit different stimulation requirements of their cognate E2s to pair with them for a specific function. Although the rate-driven pairing model described here is based on the idiosyncratic properties of Ubc6 and might therefore be specific to the ERAD system, our work sheds light into an important and largely overlooked question: Why do a substantial fraction of RING domains feature a residue at the linchpin position that is suboptimal to arginine or cannot hydrogen bond (60% in yeast—Fig 1C, 53% in humans—Fig EV1), when such an interaction is thought to be crucial for stimulating a collaborating E2? Our work suggests a picture where the E2 stimulation function of RING E3s may not be absolutely crucial to control ubiquitylation in all instances. E2s may have different stimulation requirements thus providing RINGs with the opportunity to regulate pairing with specific E2 counterparts for a desired functional outcome.

Most E2s characterized to date have conjugates that exist predominantly in open (less reactive) conformations, thereby implying linchpin sensitivity. But a majority of E2~Ub species have yet to be characterized (Stewart *et al*, 2016). Given the large fraction of RING E3s that harbor suboptimal linchpins, it is not unreasonable to propose a spectrum of E2~Ub behaviors that span from predominantly open and highly linchpin-dependent, like in the case of Ubc7 (shown here) or human UbcH5 (Pruneda *et al*, 2011) to more closed conjugates, like that of Ubc6 (shown here) or human Ube2G1 (Choi *et al*, 2015), which may show higher basal activity and be linchpin-insensitive. RING E3s will have evolved to take advantage of such differences and to "thread the needle" if their biological functions require them to pair with more than one E2. For example, Rbx1, the RING module of all CRLs, has asparagine as its linchpin in both yeast and humans. CRLs control the degradation of about a fifth of all proteasome-dependent

degradation (Soucy *et al*, 2009) raising the question as to why their RING does not provide the most efficient linchpin. Previous work showed that the asparagine linchpin in Rbx1 is actually superior to arginine in its ability to stimulate both the idiosyncratic chain building E2, Cdc34, and the functionally essential NEDD8-specific E2, Ubc12 (Scott *et al*, 2014). However, the asparagine linchpin is, as expected, less efficient than arginine at stimulating Ub transfer activity from UbcH5. CRLs have been shown to stimulate E2s of the UbcH5 family in addition to Cdc34 (Duda *et al*, 2008; Saha & Deshaies, 2008; Yamoah *et al*, 2008) and potentially use the former for priming (Wu *et al*, 2010a). But in an additional wrinkle, the priming reaction for many CRL substrates *in vivo* is carried out by the RBR (RING-between-RING) E3 ligase ARIH1 and its cognate E2 UbcH7 (Scott *et al*, 2016; Dove *et al*, 2017), which rely on a different molecular mechanism from RING E3s and their E2s (Wenzel *et al*, 2011). Analogous to the ERAD cases studied here, we speculate that the non-optimal linchpin of Rbx1 disfavors priming reactions catalyzed by Rbx1, thus promoting the preferred priming action of the ARIH1/UbcH7 pair.

Finally, we further extend our insights to another class of E3 ligases that choose from the same pool of E2s as RING E3s, specifically, the aforementioned RING-Between-RING (RBR) E3s. The idea that RINGs have evolved suboptimal linchpins for functional reasons has been invoked in the context of RBR E3 ligases. Despite containing an E2-recruiting RING domain, RBR E3s work via an E3~Ub thioester intermediate that subsequently modifies a substrate (Wenzel *et al*, 2011). Here, the RBR RING1 domain sterically disfavors closed E2~Ub conformations (Dove *et al*, 2016). Maintaining the bound E2~Ub in an open, unreactive conformation prevents premature Ub discharge from the E2 onto nearby (E3) lysine residues. Indeed, no human or yeast RBR features an arginine at the linchpin position. We expect there are variations on the themes revealed from our work and look forward to future studies that shed light on other E3 ligases and particularly on E2s to better understand their functional pairing mechanisms.

## Materials and Methods

### Plasmids, protein expression, and purification

Plasmid constructs, expression, and purification of most proteins have been previously described (see Appendix Table S1). Ubc7 containing only the catalytic cysteine ("C89only" = C39A/C141S), and the U7BR domain of Cue1 (aa 150–203) and Hrd1 (aa 325–412) were cloned into a pET28a vector containing human His$_6$-tagged SUMO3, respectively (Meulmeester *et al*, 2008); Doa10 (aa 19–102) was cloned into a pGEX-6p1 vector. Using constructs described above, S-peptide-tagged variants of all RING E3 ligases were obtained by cloning. All indicated mutants were created by site-directed mutagenesis. For expression, plasmids were transformed in BL21/DE3 bacteria and grown either in rich LB medium, TB medium, or minimal MOPS medium supplemented with ($^{15}$N)ammonium chloride. Proteins were expressed overnight at 16°C after induction at OD 0.6–0.8 with 1 mM IPTG. Recombinant proteins were enriched via affinity chromatography using Glutathione Sepharose or Ni-NTA for GST-tagged or His$_6$-tagged proteins, respectively. Depending on the construct, proteins were eluted by

imidazole or cleavage with GST-HRV-3C protease, SUMO protease (His$_6$-Ulp1 or GST-tagged, catalytic fragment of hSENP1), or thrombin (Sigma). U7BR was purified by incubation with thrombin and GST-hSENP1 after binding to Ni-NTA, followed by a washing step and imidazole elution. All proteins were subjected to Superdex 75 size exclusion chromatography as the final step of purification.

## Generation and purification of recombinant protein complexes and adducts

Labeling of Ub(S20C) with Alexa Fluor 488 C5 maleimide was previously described (Bagola *et al*, 2013). K48-linked Ub chains (His$_6$-Ub$_2$ and His$_6$-Ub$_3$) were assembled as previously described (von Delbrück *et al*, 2016). The U7BR/Ubc7(C89only) complexes used for NMR as well as the U7BR/Ubc7(wt) complexes used for *in vitro* Ub discharge assays were purified by mixing purified Ubc7 and U7BR in a 1:2 ratio and subjecting them to Superdex 75 size exclusion chromatography. Disulfide-linked E2-SS-Ub conjugates were obtained by incubating at least 200 µM of desired single-cysteine E2 variants (Ubc6 and Ubc13 naturally contain only one cysteine) with a twofold molar excess of a disulfide-linked adduct of Ub(G76C) and 2-nitro-5-thiobenzoate (TNB) for 45 min at RT followed by Superdex 75 size exclusion chromatography. The Ub(G76C)-TNB adduct was prepared prior to conjugate formation by incubating 1 mM Ub(G76C) with 10 mM DTNB (Ellman's reagent—Sigma) followed by buffer exchange over Sephadex G-25 resin.

## Isothermal calorimetry

Measurements were performed on a Microcal VP-ITC calorimeter (Malvern Panalytical) at 15°C. Proteins were dissolved in PBS (25 mM sodium phosphate pH 7.0, 150 mM NaCl). Binding information was obtained with 5 µl additions of titrant (Doa10 or Hrd1 at 3.5 mM) in 4-min intervals into 50 µM indicated E2-SS-Ub samples. Data were analyzed using Microcal PEAQ-ITC analysis software (Malvern Panalytical).

## *In vitro* substrate ubiquitylation assay

The experimental setup was modified from Bays *et al* (2001). In detail, reactions included E1 (300 nM), indicated E2 (5 µM each; for reactions that contain free Ubc7, 5 µM Cue1(wt) was added as well), indicated S-peptide-tagged RING variants (5 µM), RNase S (5 µM), and Ub (wild-type or K48R for Ubc6 and Ubc7, respectively, at 50 µM) in *in vitro* reaction buffer (50 mM Tris–HCl, pH 8.0, 4 mM MgCl$_2$, 0.5 mM DTT). Reactions were started by the addition of 4 mM ATP and incubated at 30°C for 30 min. Modification was analyzed by immunoblotting against RNase A. Signals were visualized on a Li-COR Odyssey system. Reactions were performed in triplicates. For each reaction, the RNase-Ub signal was quantified and normalized by the total signal, i.e., signal of RNase and RNase-Ub (RNase-Ub$_2$ signals were not quantified as they are often negligible and coincide with an unspecific signal).

## *In vitro* Ub discharge assays

*In vitro* Ub discharge assays were performed similar to the protocol described in Wenzel *et al* (2011). The assay allows monitoring

the kinetics of Ub transfer from a preformed E2~Ub conjugate onto a small nucleophile (Pickart & Rose, 1985; Wenzel *et al*, 2011). To enable investigation of both Ubc7 and Ubc6 activation, we chose ethanolamine as the nucleophile in light of the reported hydroxy-reactivity of Ubc6 (Wang *et al*, 2009; Weber *et al*, 2016). Similar to substrate proteins encountered by these E2s, ethanolamine provides a primary amine and a hydroxyl group as potential nucleophiles. Although the bi-functional substrate renders direct quantitative comparison between Ubc6 and Ubc7 discharge kinetics difficult due to the potentially different products formed by the two E2s, it allows for uniform assay conditions and, for a given E2, for comparison of E2 stimulation by different E3 ligases. For the assay, indicated E2s (30 µM) were charged with Ub (120 µM) in the presence of human E1 (3 µM for Ubc6 or 500 nM for Ubc7) and 5 mM ATP at 37°C for 3 min. To assay Ubc6 activity, purified Ubc6 (aa 2–179) was used. To assay Ubc7 activity, purified complex of U7BR/Ubc7 was charged with Ub(K48R) to avoid intrinsic chain formation. Charging reactions were quenched with 10 mM EDTA and, to start discharge, directly diluted 1:1 into PBS containing ethanolamine (100 mM final concentration) and indicated E3 ligases (30 µM final concentration). Reactivity time courses were performed at RT, if not indicated otherwise. Samples were quenched in non-reducing sample buffer at indicated time points and visualized by Coomassie-stained SDS–PAGE. For each time point, the band intensity for the E2~Ub conjugate and the free E2 were determined by densitometry analysis using Gel Doc EZ Imager (Bio-Rad) and the Image Lab 6.0.1 software (Bio-Rad). Discharge was quantified by plotting over time the intensity of the E2~Ub conjugate band divided by the sum of the intensities of the E2~Ub conjugate and the free E2 band. Reaction rates were derived by fitting discharge to a first-order reaction model using xcrvfit. Reactions were performed in triplicates and presented rates given as the mean of all replicates ± standard deviation.

## *In vitro* chain elongation reactions

Single-turnover chain elongation reactions included human E1 (150 nM), indicated Cue1 variants (2 µM), free Ubc7 (2 µM), indicated E3 ligase variants (2 µM), acceptor Ub (C-terminally His$_6$-blocked mono-Ub or indicated K48-linked Ub chains at 14.8 µM each), and donor Ub (fluorescent Ub(S20C)-C5 at 200 nM) in *in vitro* reaction buffer. Reactions were started by the addition of 4 mM ATP and incubated at 30°C. Samples were quenched in reducing sample buffer after 0, 1, 2, 3, 4, 5, 10, 30, and 60 min and visualized by SDS–PAGE. Product formation over time was quantified by fluorescence at 488 nm using a Typhoon FLA9500 (GE Healthcare) and ImageQuant TL 7.01 software (GE Healthcare). Reaction rates were derived by fitting discharge to a first-order reaction model using GraphPad Prism 6 software. Reactions were performed in triplicates and presented rates given as the mean of all replicates ± standard deviation.

## Yeast strains

All strains are haploid descendants of DF5 (trp1-1 (am)/trp1-1 (am), his3-Δ200/his3-Δ200, ura3-52/ura3-52, lys2-801/lys2-801, leu2-3, -112/leu2-3, -112, MATa/α) (Finley *et al*, 1987). Strains were generated by standard transformation or crossing protocols.

Appendix Tables S2 and S3 list the yeast expression plasmids and yeast strains, respectively, used in this study. Appendix Fig S5 shows a validation of yeast strains.

### Cycloheximide decay assay

Cycloheximide decay assays were performed as previously described (Weber *et al*, 2016). In brief, log-phase growing yeast cells were resuspended in cycloheximide-containing SD medium (Sigma, final concentration: 0.33 mg/ml). At indicated time points, samples were collected and degradation was stopped by the addition of 10 mM sodium azide. Cell lysates were prepared by the addition of glass beads and vigorous shaking. Samples were analyzed by immunoblotting. Appendix Table S4 lists the antibodies used for immunoblotting. For means of quantification, fluorescently labeled secondary antibodies were used and visualized on a Li-COR Odyssey system. Experiments were performed in triplicates.

### Pulse-chase assay

Pulse-chase assays were performed as previously described (Mehnert *et al*, 2014). In brief, log-phase growing yeast cells were resuspended in SD medium. Newly synthesized proteins were labeled with radioisotopes by the addition of 3 MBq *EasyTag™ EXPRESS$^{35}$S Protein Labeling Mix* (PerkinElmer, Inc.) and incubated at 30°C for 8 min. Cells were resuspended in label-free chase mix (SD medium, 3.3 mM ammonium sulfate, 0.013% (*w/v*) L-methionine, 0.01% (*w/v*) L-cysteine) and incubated at 30°C. At indicated time points, samples were collected and degradation was stopped by the addition of 10 mM sodium azide. Cell lysates were prepared by the addition of glass beads and vigorous shaking. The protein of interest was immunoprecipitated with specific antibodies (see Appendix Table S4). Samples containing PrA*-3xHA or CPY*_K0-HA were treated with endoglycosidase F. Radioactive signals were analyzed by autoradiography using a Typhoon FLA9500 (GE Healthcare) and ImageQuant TL 7.01 software (GE Healthcare). Experiments were performed at least in triplicates.

During monitoring of Hrd1 model substrate degradation by CHX decay, we experienced technical problems (signal detection/data quality needed for quantification). Hence, we switched to pulse-chase experiments for these substrates. Since sample preparation in pulse-chase assays is inherently more error-prone (immunoprecipitation of substrate out of every sample), we tried to reduce the use of this assay by analyzing Doa10 model substrate degradation via CHX analysis.

### NMR spectroscopy

NMR spectra were recorded on a 500 MHz Bruker Avance II (University of Washington) at 22°C in 25 mM sodium phosphate pH 7.0, 150 mM NaCl, 10% D2O. $^{1}$H/$^{15}$N-HSQC-TROSY experiments were collected on 200 μM labeled protein, if not indicated otherwise. Datasets were processed using NMRPipe/NMRDraw (Delaglio *et al*, 1995) and visualized with NMRView (Johnson & Blevins, 1994). Chemical shift perturbations observed by 2D HSQC-TROSY NMR were quantified in parts per million with the equation $\Delta\delta$ $(^{1}H/^{15}N) = \{[\delta(^{1}H) - \delta(^{1}H)_0]^2 + 0.04[\delta(^{15}N) - \delta(^{15}N)_0]^2\}^{1/2}$.

Nitrogen relaxation time analyses were performed with delays of 10, 40, 80, 120, 160, 320, 640, and 1,000 ms for T1 and 8.48, 33.92, 42.4, 25.44, 59.36, 16.96, 50.8, and 67.84 ms for T2. Rates and their respective errors for each analyzed residue were quantified using the built-in "rate analysis" function of NMRView. Average T1/T2 ratios for the ($^{15}$N)Ub(G76C) constructs shown in Fig 5C are presented as the respective means ± standard deviation for all analyzed residues.

## Data availability

This study includes no data deposited in external repositories.

**Expanded View** for this article is available online.

## Acknowledgements

We thank the Rapoport group for their generous gift of the CPY*-K0-HA yeast expression plasmid, Damien Wilbur for help with NMR analysis, Corinna Volkwein for technical support with pulse-chase assays, and Peter S. Brzovic, Oliver Daumke, and Alba M. Ferri Blazquez, Ernst Jarosch, Achim Werner, Jonathan N. Pruneda, Mikaela Stewart for critical reading of the manuscript. This work was generously supported by the German Research Foundation DFG (SPP 1365 and SFB 740/TP B05 to TS; postdoctoral fellowship RID. 2874/1-1 to TR) and the National Institute of General Medical Sciences (Grant R01 GM088055 to REK).

## Author contribution

CL and TR carried out the experiments with support from MM and MKJ. REK and TR wrote the manuscript with support from CL, AW, MKJ, and TS. REK and TS supervised the project. All authors provided critical feedback and helped shape the research, analysis, and manuscript.

## Conflict of interest

The authors declare that they have no conflict of interest.

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
