## [Review Process File · The EMBO Journal]

Who with whom: Functional coordination of E2 enzymes by RING E3 ligases during poly-ubiquitylation

Christian Lips, Tobias Ritterhoff, Annika Weber, Maria Janowska, Mandy Mustroph, Thomas Sommer, and Rachel Klevit

DOI:10.15252/embj.2020104863

Corresponding author(s): Rachel Klevit (klevit@u.washington.edu)

Review Timeline:	Submission Date:	1st Mar 20
	Editorial Decision:	15th Apr 20
	Revision Received:	28th Jul 20
	Editorial Decision:	20th Aug 20
	Revision Received:	31st Aug 20
	Accepted:	10th Sep 20

Editor: Hartmut Vodermaier

Transaction Report:

Thank you again for submitting your full manuscript on ERAD E2-E3 coordination, and apologies for the delay in getting back with a decision - as we are currently experiencing an unusually high number of submissions, and as I was also out of office for several days over the Easter holidays. We have in the meantime received a complete set of reports, and also been able to circulate it between the three reviewers and discuss the comments within our team. As you will see from the reports copied below, the referees remain presently somewhat divided in their opinions: While referee 2 appreciates the interest to the field and acknowledges the overall quality of the work, both referees 1 and 3 raise a number of technical and interpretational concerns that would require decisive clarification, at least in part via additional experimentation (even though we do not feel that a structural model would be essential at this point). Moreover, the reports indicate that it is not easy to derive more general conclusions on RING E3-E2 cooperation (e.g. the concept of 'activity selection' being not clearly worked out) at present, and that there is insufficient discussion of relevant literature by other groups.

Given that the studied E2s and E3 represent a widely-studied paradigm, we would still remain open to considering a revised manuscript further, should you feel you might be able to satisfactorily address these issues and provide more definitive support for the conclusions and interpretations. It is however clear that convincing the referees will likely require substantial further time and efforts, preventing me from making strong commitments regarding eventual EMBO Journal publication at this stage. I also realize that the current COVID-19 crisis situation and associated difficulties with lab access and experimental work may not allow comprehensive revision of all critical issues in a timely manner, in which case resubmission at a later point as well as publication of a less extensive revision in a sister journal (e.g. EMBO reports) would be options we could discuss. Please do not hesitate to get back to me once you have considered the reports in depth together with your collaborators, I would be happy to talk about the comments and revision possibilities also directly with you via phone or Skype.

Referee #1:

Ubiquitin (Ub) E3 ligases can use distinct E2 enzymes for substrate priming and Ub chain elongation. Yet, how E3s select E2 enzymes for these reaction steps is largely unknown. In the yeast ERAD pathway, the Hrd1 RING ligase uses a single E2, Ubc7, for priming and chain elongation, while the Doa10 RING E3 cooperates first with the dedicated priming E2 Ubc6 and then with Ubc7 for chain elongation.

Using a combination of NMR spectroscopy, biochemical and in vivo studies, Lips et al. shed light on the mechanisms of E2 selection by Hrd1 and Doa10. The authors suggest that for both E3s the priming reaction is the rate-limiting step for substrate degradation. The authors show that Ubc7 rather than Ubc6 functions as priming E2 for Hrd1 based on its higher affinity to the RING domain. For priming, Ubc7 requires allosteric activation by a so-called linchpin residue in the Hrd1 RING domain, since the Ubc7~Ub intermediate alone seems to adopt an open, inactive conformation. In contrast, Doa10 interacts with Ubc6 and Ubc7 with equally low affinity and contains an atypical linchpin residue that cannot activate Ubc7. However, since the Ubc6~Ub intermediate may already be biased towards a closed, active conformation prior to E3 binding, its priming activity is independent of E3-mediated activation. The authors therefore suggest that based in its higher intrinsic activity Doa10 selects Ubc6 over Ubc7 as priming E2. Lastly, the authors provide evidence that, unlike in the priming reaction, the Ubc7 co-factor Cue1 enhances chain elongation more efficiently than E3-mediated E2 activation explaining why both E3s collaborate with the Ubc7/Cue complex during poly-Ub chain synthesis.

While E2/E3 selection is an important and interesting subject, the conclusions of the authors remain largely suggestive and are not sufficiently supported by the data. Moreover, the manuscript is difficult to read and interpret since key aspects such as the linchpin residue are not well introduced and assays are not performed in a consistent and sufficiently detailed manner (see below). With that said, I feel that the quality of the manuscript does not meet the standards for publication in EMBO Journal.

Specific issues:

- 1) The linchpin residue as well as the relation between the open and closed E2~Ub conformations should be introduced in more detail in the Introduction. It would be helpful to visualize the linchpin residue in a known E2~Ub:RING structure and show a sequence alignment of the Hrd1 and Doa10 RING domain highlighting the linchpin residue.
- 2) The concept of "activity selection" is unclear to me. How can a higher intrinsic activity of the Ubc6~Ub thioester be a selection criterion for E2 recognition? Also, the affinities of the E2~Ub thioester to the RING domain should matter for E2 selection and not the affinity of the E2 alone. As such the data may explain why priming is more efficient for Doa10 with Ubc6 than with Ubc7, but they do not provide a basis for E2 selection.
- 3) The rates of E2~Ub discharge in the presence of the E3s are extremely slow. A rate of ~30 discharged Ub moieties per hour translates into one Ub transferred every two minutes. This can hardly be physiologically relevant, as in fact the observed in vivo degradation rates would suggest. Moreover, for all discharge assays representative gels showing the analyzed bands and the decay curves should be shown. The same holds for the CHX decay assays.

- 4) Ubc6 is more active with an Arg linchpin. This demonstrates that the linchpin residue not only plays a role for Ubc7, but also for Ubc6. Along the same line, the results in Fig. 3D and E are inconsistent. Why would Ub discharge activity be affected by the linchpin, but not priming?
- 5) Why is substrate degradation for Hrd1 and Doa10 not assayed in the same way (pulse-chase for Hrd1 versus CHX decay for Doa10)?
- 6) How comparable is Ub discharge in the presence of ethanolamine if Ubc6 is hydroxy-reactive but not Ubc7?
- 7) CSPs alone do not provide information on the conformation of the E2~Ub thioester. High-resolution structures or at least data-based models would be required to report on the conformation of the E2~Ub thioester.
- 8) To investigate whether Cue1 can indeed overrule the dependence of E2 activity on the linchpin residue, the experiments would have to be performed in a Cue1 concentration dependent manner.
- 9) Since reaction rates are highly sensitive to reactant concentrations, the observed discrepancies between in vitro and in vivo data may simply reflect concentration differences and thus should be interpreted with caution.

Referee #2:

Ubiquitin chain formation requires chain initiation, i.e. modification of a substrate residue, and chain elongation, i.e. modification of ubiquitin residues. How E3 ligases select their cognate E2 enzymes, activate them, and implement them at each step of chain formation is still incompletely understood. Revealing mechanisms of ubiquitin chain formation has important implications in our understanding of cellular signal transduction in development and disease.

In this collaborative paper, the Klevit and Sommer labs investigate the interactions between two well-established ERAD E3 ligases, DOA10 and HRD1, and the ERAD E2s UBC6 and UBC7. They show that Hrd1 basically selects its E2 based on affinity, as it prefers to bind UBC7 over UBC6. This includes the identification of UBC7-dependent chain initiation. By contrast, DOA10, which contains a sub-optimal linchpin residue, can activate the linchpin-insensitive UBC6, while it is much less efficient in activating linchpin-dependent UBC7. UBC6 is less dependent on the linchpin residue, as it populates the closed E2~Ub conformation more frequently than UBC7 and can therefore transfer its ubiquitin to a nucleophile even in the absence of an E3. They conclude their study by showing that under sensitized conditions, each E3 can prime chain formation with their non-optimal E2 as well.

The experiments in this study have been executed well and are discussed carefully. Albeit somewhat specialized, this work does provide some insight into E2 selection by E3 ligases that extends beyond what had been published previously. I have no major issues that absolutely require experiments, and I believe that this paper will after some rewriting be interesting for researchers in the ubiquitin community.

Major points:

1. The NMR experiments in Figure 1 suggest that Ubc6 binds both Hrd1 and Doa10 with similar affinities in the mM range. However, the authors should state more explicitly that this applies only to the isolated UBC domain. The full-length E2 contains a transmembrane domain and might be presented differently to each E3s. This is a caveat of these measurements that needs to be pointed out clearly (I am hesitant to ask for experiments revealing the similar affinity of full-length UBC6 to both E3s, given the current situation).

2. I am concerned about the discrepancy between the in vitro and in vivo experiments in Figure 2. The most trivial explanation is that the in vitro setup does not fully reproduce the in vivo situation. Alternatively, however, as discussed later in the article, there could be another rate-limiting step than polyubiquitylation, i.e. chain initiation or proteasomal shuttling. If possible, it would therefore be helpful to monitor ubiquitylation, rather than degradation, of model substrates in the different chain backgrounds (I acknowledge that the data in Fig. 3 strongly points to chain initiation as the rate-limiting step, so this could also be addressed by re-writing this part of the paper to avoid confusion).

Minor points:

1. When mentioning the close E2~Ub conformation in the introduction, the authors should cite the two first papers showing it: Saha et al., Mol Cell 2011 and Wickliffe et al., Cell 2011.

2. The statement that little is known about physiological E3s that use it for priming is not correct. Recent work by Kleiger and colleagues, as well as by the Schulman lab showed it for SCF, and in vivo data for the APC/C has been reported as well. This should be cited more accurately.

Referee #3:

How E3 and E2 ligases cooperate and select each other is an important question that has been characterized for selected model systems. This manuscript analyzes the E2-E3 interplay for the yeast ERAD RING E3 ligases Hrd1 and Doa10 with Ubc6 and 7, and combines in vitro and cellular assays with NMR. It has previously been shown that Hrd1 works with Ubc7 in vivo while Doa10 uses Ubc6 and 7 for priming and elongation, respectively. In the present manuscript, the authors use kinetic analyses to conclude that Hrd1 uses Ubc7 for priming and that priming is the rate-limiting step in protein degradation directed by Hrd1 and Doa10. The abstract also claims that the study offers "a mechanistic framework to understand functional selection and coordination of E2 enzymes by RING E3 ligases in general". However, the paper falls short of showing evidence that the findings may be generalised. I feel that this point needs to be worked out more to add impact, in addition to the issues below, many of which are technical.

MAJOR CONCERNS:

Figure 1C

To compare the substrate ubiquitination activities of Hrd1 in the presence of UBC6 and 7, it is necessary to show these reactions on the same Western blot. The Western should be repeated and quantified.

As is, the data does not look convincing, since the right blot seems less exposed than the left one

(see much darker shadow underneath the RNase band - what is this shadow anyway?).

Figure 1D

The actual data appear to be missing. Those should be presented along with the quantification.

Figure 1 E, F:

It remains unclear how and why "selected residues" were chosen. It may be more suitable to define some cut-off and include all residues within this range for K_D analysis. This should also be done in the analyses in Figure 2S, because it may illustrate mutation induced differences in binding mode. If any, those should be discussed.

Page 7:

The authors state that "The linchpin mutations do not substantially alter E2/E3 binding affinity". To be able to conclude this, the NMR titration data should be fitted and K_D values reported. In Figure S2G, it looks like the critical arginine mutation of the linchpin in Doa1 does have an effect, but without a fit, it is hard to say. For the fitting it would be important to use the same data range for mutants and WT (the same maximum concentration of ligand). To also account for possible changes in the binding mode, the authors should analyze all resonances within a certain cut-off (see comment above) and not 4 selected ones.

Figure 2A, B:

The data from which the rates were derived should be shown in the supplements.

Page 8:

The authors conclude "The simplest explanation for the disparity between the in vitro elongation kinetics and in vivo substrate degradation is that chain elongation is not rate-limiting to the process of protein degradation in yeast. This implies that the "minimal E3-stimulated elongation rates" observed in the in vitro assays with wild-type-RINGs (see Fig. 2B) are sufficient to guarantee effective protein degradation at both E3s under the conditions assayed in vivo."

The first sentence makes sense. The second one seems overstated. There are many factors in the cell that may alter the rates compared to the in vitro situation...

Figure 2C:

Protein degradation is monitored by pulse-chase experiments for the Hrd1 model substrate and by CHX decay assays for the Doa10 model substrate? The authors should give an explanation as to why two types of assays are necessary.

Figure 3A:

The errors look quite large, making it necessary to demonstrate significance.

Figure 3D:

The authors state "the differences among linchpin mutations were far less pronounced than for Ubc7 (compare Figure 2A)"

However, it seems that the difference between WT and R is about 3 fold and the same is seen in 2A.

Figure 3E:

To compare the linchpin dependencies of Doa10 with Ubc6 and 7, they should be on the same blot.

Figure 4A:

This assay lacks a control (time point zero, minus ATP...) and +DTT samples to provide evidence for the thioester versus autoubiquitinated species (important for Ubc7~Ub in particular). Representative data for the reactions with nucleophile appear to be missing.

Figure 5:

Representative raw data appear to be missing. Likewise controls, e.g. experiments illustrating the identity of the yeast strains.

Figure S4B/D:

The RGA mutation makes an obvious difference here. However, the authors state "the Cue1(RGA) mutation that is so detrimental to in vitro elongation rates had virtually no effect on substrate degradation kinetics for either E3 (Fig. 2C and Fig. S4A+B - compare solid vs. dashed traces in each panel)"

Also, the authors state that "virtually no difference in the degradation rates of Doa10 substrates was observed (Fig. 3C and Fig. S4D). However, there is a clear differences between the mutants in S4D.

The authors should explain their interpretation and/or make it more precise.

Figure S5:

Why is the signal in the top blot so much stronger on the left than on the right? It makes it hard to interpret.

MINOR CONCERNS:

Page 4:

The authors state "There are limited examples of RING E3s that are confirmed to collaborate with a priming and an elongation E2 to catalyze attachment of poly-Ub chains to substrates(Parker and Ulrich, 2009; Rodrigo-Brenni and Morgan, 2007; Weber et al., 2016; Wu et al., 2010). Therefore, understanding of the specific role of E3 ligases and their interplay with E2 enzymes during the steps of poly-ubiquitylation is still rather limited. "

This overlooks landmark work by Schulman, Rape and others that has provided exquisite detail of the roles of priming and elongating E2s in the context of the APC. This work should be included in the introduction.

When introducing the closed conformation (page 3), the authors should cite Deshaies and Rape (Saha et al., Mol Cell 2011 and Wickliffe et al., Cell 2011).

Page 5:

The authors show by NMR that Hrd1 has significantly different affinities for Ubc6 and 7, with the latter interaction only in the millimolar range. Doa10 binds both E2s only in the millimolar range (yet works with both E2s). This suggests, indeed, that Hrd1 may use affinity to discriminate the E2s. However, it also shows that the in vitro affinities only partially reflect what is happening in the cell (since Doa10 works with both E2s, despite both being in the millimolar affinity range in vitro).

The authors should quote the KD values in the text and comment on this point, since the situation is more complex than the current interpretation indicates.

Figure 1A:

The table needs some streamlining. It is meant to list the functions of each protein, but lists tagged

substrates and general properties.

Figure S6C:

One of the blue data needs recoloring, since they are indistinguishable in the legend.

Figure S1D:

The highlighted residues should be labeled on the structures.

Figure S1G-H:

Aren't there NMR assignments of yeast Hrd1 that could be transferred here?

RESPONSE TO REVIEWERS

General Comments.

We thank reviewers for their constructive criticism and have revised the manuscript to address and clarify. The revised manuscript is substantially changed, mainly in the Introduction and the Discussion sections, with major changes outlined below. Point-by-point responses follow.

To respond to requests for a clearer and more thorough presentation of current knowledge and models, we provide a (new) figure in the Introduction that summarizes the concepts and components important to the manuscript. This led to a change of figure numbering – in our responses, we will use the current numbering unless stated otherwise.

Rather than reporting binding affinities of E2/E3 interactions (formally Fig. 1B and Fig. S2), we now report the affinities between the E3s and E2~Ub conjugate mimics, as these are more relevant (Fig. 2A). The values now reported were determined by isothermal titration calorimetry (ITC) rather than NMR as in the original manuscript.

We use nucleophile discharge assays to assess the stimulatory effect of E3s on the ability of a given E2~Ub conjugate to discharge Ub. Comparing these across different E2s is problematic and we have avoided doing this. To stress this point more clearly, we now report the results of discharge assays as “fold stimulation” of an E3 for a given E2, calculated as the ratio of discharge rate of the E2 in the presence of an E3 and that of the “no E3” control. Absolute discharge rates, as presented in the original submission, are now provided in the Expanded View (Fig. EV2C-F). During our re-analysis, we discovered an error in one of four replicates previously used in the Ubc7 data. The corrected data, now included, differ only nominally from the data originally reported.

Lastly, we have greatly reshaped the discussion in response to reviewers' comments. In particular, we now highlight why and how our story has broader relevance than just to the ERAD system investigated. It is not generally appreciated that over half of all known RING E3s have suboptimal linchpin residues and therefore may not be accurately understood in terms of the canonical linchpin mechanism of activation. Furthermore, the criteria that are important for how a given E3 “chooses” its functional E2(s) have not been defined. Our results and interpretation provide novel insights for both of these fundamental unanswered issues in protein ubiquitylation.

Specific (point-by-point) Comments.

Referee #1:

Ubiquitin (Ub) E3 ligases can use distinct E2 enzymes for substrate priming and Ub chain elongation. Yet, how E3s select E2 enzymes for these reaction steps is largely unknown. In the yeast ERAD pathway, the Hrd1 RING ligase uses a single E2, Ubc7, for priming and chain elongation, while the Doa10 RING E3 cooperates first with the dedicated priming E2 Ubc6 and then with Ubc7 for chain elongation.

Using a combination of NMR spectroscopy, biochemical and in vivo studies, Lips et al. shed light on the mechanisms of E2 selection by Hrd1 and Doa10. The authors suggest that for both E3s the priming reaction is the rate-limiting step for substrate degradation. The authors show that Ubc7 rather than Ubc6 functions as priming E2 for Hrd1 based on its higher affinity to the RING domain. For priming, Ubc7 requires allosteric activation by a so-called linchpin residue in the Hrd1 RING domain, since the Ubc7~Ub intermediate alone seems to adopt an open, inactive conformation. In contrast, Doa10 interacts with Ubc6 and Ubc7 with equally low affinity and contains an atypical linchpin residue that cannot activate Ubc7. However, since the Ubc6~Ub intermediate may already be biased towards a

closed, active conformation prior to E3 binding, its priming activity is independent of E3-mediated activation. The authors therefore suggest that based in its higher intrinsic activity Doa10 selects Ubc6 over Ubc7 as priming E2. Lastly, the authors provide evidence that, unlike in the priming reaction, the Ubc7 co-factor Cue1 enhances chain elongation more efficiently than E3-mediated E2 activation explaining why both E3s collaborate with the Ubc7/Cue complex during poly-Ub chain synthesis.

While E2/E3 selection is an important and interesting subject, the conclusions of the authors remain largely suggestive and are not sufficiently supported by the data. Moreover, the manuscript is difficult to read and interpret since key aspects such as the linchpin residue are not well introduced and assays are not performed in a consistent and sufficiently detailed manner (see below). With that said, I feel that the quality of the manuscript does not meet the standards for publication in EMBO Journal.

We are pleased that the reviewer recognizes the importance of the subject. We have attempted to address the identified shortcomings in our revised manuscript. We added data and text to more thoroughly present our rationales and to be more precise in our conclusions. We have strengthened specific points and streamlined the discussion towards the importance of linchpin-mediated E2 activation by RING E3 ligases. We believe these changes clarify the impact of our study as about half of all RING E3s feature a non-canonical linchpin residue. We acknowledge that the manuscript is dense and hope that the revisions allow for easier reading.

Specific issues:

1) The linchpin residue as well as the relation between the open and closed E2~Ub conformations should be introduced in more detail in the Introduction. It would be helpful to visualize the linchpin residue in a known E2~Ub:RING structure and show a sequence alignment of the Hrd1 and Doa10 RING domain highlighting the linchpin residue.

We have provided a more thorough description of linchpin-mediated E2 activation to the introduction and dedicate a new figure (Fig. 1) that includes the requested information.

2) The concept of "activity selection" is unclear to me. How can a higher intrinsic activity of the Ubc6~Ub thioester be a selection criterion for E2 recognition? Also, the affinities of the E2~Ub thioester to the RING domain should matter for E2 selection and not the affinity of the E2 alone. As such the data may explain why priming is more efficient for Doa10 with Ubc6 than with Ubc7, but they do not provide a basis for E2 selection.

We can see how the term "activity selection" might not adequately reflect our concept. Upon further consideration, we propose the term "rate-driven pairing" for Doa10's utilization of Ubc6 during priming. We hope that this term better conveys our idea that the higher intrinsic discharge activity of Ubc6 enables Doa10 to use Ubc6 during priming based in its ability to recruit the Ubc6~Ub thioester and the substrate and *not* on its ability to stimulate discharge from the E2). In contrast, Ubc7 absolutely requires E3-mediated stimulation for efficient Ub discharge. Therefore, it is unlikely that the interaction of Doa10 with Ubc7~Ub results in functional priming event (as shown in Fig. 4D).

We agree that the affinities of the RING domains for the free E2s do not adequately represent the situation during functional pairing and have replaced these with new measurements (see General Comments on first page of this document).

3) The rates of E2~Ub discharge in the presence of the E3s are extremely slow. A rate of ~30 discharged Ub moieties per hour translates into one Ub transferred every two minutes. This can hardly be physiologically relevant, as in fact the observed in vivo degradation rates would suggest.

Moreover, for all discharge assays representative gels showing the analyzed bands and the decay curves should be shown. The same holds for the CHX decay assays.

Indeed, discharge rates are slow and do not reflect rates of physiological ubiquitylation events. Discharge assays are not meant to simulate the kinetics of Ub transfer onto a specific target; in that reaction, the E3 binds and presents substrate as well as binding and stimulating E2~Ub. Substrate proximity provides a powerful rate enhancement. We use discharge assays as a tool to assess the relative stimulatory effect of E3s on the ability of a given E2~Ub conjugate to discharge Ub onto a nucleophile (which is present at huge excess to make the reaction zero-order in nucleophile). Hence, absolute rates of discharge assays ought not be overinterpreted.

Representative gels/blots/autoradiographs/fluorescence scans and decay curves for all presented experiments appear in supplemental figures. Representative gels and decay curves have been added to Fig. 2C and D) to demonstrate the workflow used to obtain the “fold stimulation” values in our presentation.

4) Ubc6 is more active with an Arg linchpin. This demonstrates that the linchpin residue not only plays a role for Ubc7, but also for Ubc6. Along the same line, the results in Fig. 3D and E are inconsistent. Why would Ub discharge activity be affected by the linchpin, but not priming?

Our point is not that Ubc6 is completely insensitive to the action of a linchpin, but rather that, due to its high basal activity, it is not utterly dependent on it for Ub transfer, unlike Ubc7. As the reviewer points out, results from discharge assays with the Doa10 linchpin variant imply that Ubc6 can be stimulated when an optimal Arg linchpin is present. However, we would like to point to results from discharge experiments with Hrd1 linchpin variants (Fig. EV2F+G) where the Arg linchpin does NOT result in drastically increased stimulation of Ubc6 discharge compared to the other linchpin variants. Moreover, the hyper-active effect of the Doa10(94R) mutant is not recapitulated in *in vitro* substrate ubiquitylation assays (Fig. 4D) nor does it result in faster degradation of Doa10 model substrates (Fig. 4A – right panel), as the reviewer points out. The disparity likely stems from the inherent differences of the methods and speaks to our conclusion that, in the case of Ubc6, the E3 function of substrate recruitment, rather than of Ub transfer stimulation, is the more important one.

5) Why is substrate degradation for Hrd1 and Doa10 not assayed in the same way (pulse-chase for Hrd1 versus CHX decay for Doa10)?

Originally, we tried to monitor degradation of all substrates by CHX decay assays but we experienced technical difficulties (detection/data quality needed for quantification) for the Hrd1 substrates. We switched to pulse-chase experiments for these substrates. Sample preparation in pulse-chase assays is technically more variable (immunoprecipitation of substrate for every sample), so we tried to limit the use of this method and therefore kept the CHX analysis for Doa10 substrates.

6) How comparable is Ub discharge in the presence of ethanolamine if Ubc6 is hydroxy-reactive but not Ubc7?

We are careful NOT to compare discharge activities of different E2s and clearly state so in the revised manuscript. This is, as the reviewer alluded to, because the nucleophile chosen for our assays, ethanolamine, is bi-functional as dictated by the differing reaction preferences (amino- versus hydroxyl- groups) of the two E2~Ubs under investigation. We used ethanolamine to allow uniform assay conditions rather than to provide the same nucleophilic group. Instead of comparing the activities of different E2s, we use discharge assays to compare the stimulatory effects of RING E3s and their mutants for a given E2. An exception to this is the assay

presented in Fig. 5A where direct comparison of the hydrolytic stability of the Ubc7~Ub and Ubc6~Ub conjugate is reasonable, as the reacting nucleophile, i.e. water, is the same for both E2s.

7) CSPs alone do not provide information on the conformation of the E2~Ub thioester. High-resolution structures or at least data-based models would be required to report on the conformation of the E2~Ub thioester.

The use of CSPs as a metric of the closed conformation is well-vetted and based on numerous experimental studies of a variety of E2~Ub conjugates (Choi et al., 2015; Dove et al., 2016; Pruneda et al., 2012; 2011; Wickliffe et al., 2011). We now provide ¹⁵N T1 and T2 relaxation measurements that report on the molecular tumbling time of Ub in different conjugates (see Fig. 5C and Appendix Fig. S18). These data unambiguously show that the Ubc6-SS-Ub conjugate behaves more like a single molecular entity compared to the U7BR/Ubc7-SS-Ub and Ubc13-SS-Ub conjugates, where the Ub moiety behaves more independently. The combination of very large CSPs around Ub's hydrophobic patch, the decreased molecular tumbling time of Ub, and previously published data about E2~Ub closed conformations all support our conclusion about the Ubc6~Ub conjugate and meet the standard of the field.

8) To investigate whether Cue1 can indeed overrule the dependence of E2 activity on the linchpin residue, the experiments would have to be performed in a Cue1 concentration dependent manner.

In the revised manuscript, we have rephrased this section to make more precise conclusions (pages 10-12). The main points we hope to convey are 1) that the Cue1(RGA) mutation decreases rates of all elongation steps to the level of the first reaction (mono-Ub to di-Ub) and 2) that for elongation reactions beyond di-Ub, a defect caused by a RING linchpin mutation in the presence of wild-type Cue1 is smaller or equal to that of Cue1(RGA) in the presence of wild-type RINGs. We acknowledge that the effects we observed may nominally be dependent on protein concentrations in the assay (particularly for the E3s). Nevertheless, our conclusions are valid for the data we provide.

There are conceptual issues with performing elongation assays in a Cue1 concentration-dependent manner. The Ubc7/Cue1 interaction is very tight (Biederer et al., 1997; Kostova et al., 2009; Metzger et al., 2013) so the Ubc7/Cue1 complex forms quantitatively under the conditions of the assays. Moreover, the interaction of Ubc7 with the U7BR domain of Cue1 allosterically activates the E2 for Ub transfer (Bazirgan and Hampton, 2008; Kostova et al., 2009; Metzger et al., 2013). In Cue1 concentration-dependent elongation assays we would thus expect to see an increase in rates at substoichiometric Cue1 levels but this would simply reflect the fact that unbound Ubc7 is inactive. Similarly, we expect to see reaction rates plateau after titration of equimolar amounts of Cue1. As an added layer of complexity, U7BR-binding of Ubc7 has been shown to increase the affinity of the E2 for Hrd1 itself (Metzger et al., 2013). This would make interpretations from the Cue1 concentration dependent experiment more difficult.

9) Since reaction rates are highly sensitive to reactant concentrations, the observed discrepancies between *in vitro* and *in vivo* data may simply reflect concentration differences and thus should be interpreted with caution.

We appreciate that reaction rates are sensitive to reactant concentrations. We do not report any reactions that are performed both *in vitro* and *in vivo*. Importantly, we are unable to follow the rate of Ub transfer to substrate *in vivo* but rather follow a later step in the process, substrate degradation. Therefore, reaction rates cannot be compared among the different types of assays. We have added a schematic in Figure 3C to clarify this situation. In the revised manuscript we have attempted to make clear what each type of assay (reaction) reports on.

Referee #2:

Ubiquitin chain formation requires chain initiation, i.e. modification of a substrate residue, and chain elongation, i.e. modification of ubiquitin residues. How E3 ligases select their cognate E2 enzymes, activate them, and implement them at each step of chain formation is still incompletely understood. Revealing mechanisms of ubiquitin chain formation has important implications in our understanding of cellular signal transduction in development and disease.

In this collaborative paper, the Kleivit and Sommer labs investigate the interactions between two well-established ERAD E3 ligases, DOA10 and HRD1, and the ERAD E2s UBC6 and UBC7. They show that Hrd1 basically selects its E2 based on affinity, as it prefers to bind UBC7 over UBC6. This includes the identification of UBC7-dependent chain initiation. By contrast, DOA10, which contains a sub-optimal linchpin residue, can activate the linchpin-insensitive UBC6, while it is much less efficient in activating linchpin-dependent UBC7. UBC6 is less dependent on the linchpin residue, as it populates the closed E2~Ub conformation more frequently than UBC7 and can therefore transfer its ubiquitin to a nucleophile even in the absence of an E3. They conclude their study by showing that under sensitized conditions, each E3 can prime chain formation with their non-optimal E2 as well.

The experiments in this study have been executed well and are discussed carefully. Albeit somewhat specialized, this work does provide some insight into E2 selection by E3 ligases that extends beyond what had been published previously. I have no major issues that absolutely require experiments, and I believe that this paper will after some rewriting be interesting for researchers in the ubiquitin community.

We thank the reviewer for the positive feedback and for supporting the publication of our work.

Major points:

1. The NMR experiments in Figure 1 suggest that Ubc6 binds both Hrd1 and Doa10 with similar affinities in the mM range. However, the authors should state more explicitly that this applies only to the isolated UBC domain. The full-length E2 contains a transmembrane domain and might be presented differently to each E3s. This is a caveat of these measurements that needs to be pointed out clearly (I am hesitant to ask for experiments revealing the similar affinity of full-length UBC6 to both E3s, given the current situation).

We appreciate that the situation in the cell is more complex and have added a paragraph (pages 6) to explicitly state that our data refer only to the soluble parts of the ERAD ubiquitin enzymes. Moreover, we show in the revised manuscript that the E2~Ub conjugates have higher affinities (μM K_D values) to the RINGs than the corresponding free E2s and acknowledge that tethering of all components to the ER membrane may lead to higher local concentrations.

It is not trivial to express full-length Ubc6 (including its transmembrane domain) recombinantly due to solubility issues. In our hands, a Ubc6 construct in which only the transmembrane domain has been deleted, behaves functionally (*in vitro*) just like the construct we used in the manuscript.

2. I am concerned about the discrepancy between the *in vitro* and *in vivo* experiments in Figure 2. The most trivial explanation is that the *in vitro* setup does not fully reproduce the *in vivo* situation. Alternatively, however, as discussed later in the article, there could be another rate-limiting step than polyubiquitylation, i.e. chain initiation or proteasomal shuttling. If possible, it would therefore be helpful to monitor ubiquitylation, rather than degradation, of model substrates in the different chain

backgrounds (I acknowledge that the data in Fig. 3 strongly points to chain initiation as the rate-limiting step, so this could also be addressed by re-writing this part of the paper to avoid confusion).

Unfortunately, we were unable to monitor ubiquitylation of the presented substrates due to technical difficulties. We acknowledge that substrate degradation is far more complicated and involves a lot more players than those involved in substrate ubiquitylation. In addition to thorough rewriting of this section (pages 8-11), we added an overview reaction panel (Fig. 3C) to better present the complexity of the process of ERAD substrate degradation and the window our experiments report on.

Minor points:

1. When mentioning the close E2~Ub conformation in the introduction, the authors should cite the two first papers showing it: Saha et al., Mol Cell 2011 and Wickliffe et al., Cell 2011.

These references have been added.

2. The statement that little is known about physiological E3s that use it for priming is not correct. Recent work by Kleiger and colleagues, as well as by the Schulman lab showed it for SCF, and in vivo data for the APC/C has been reported as well. This should be cited more accurately.

We have rephrased this statement more carefully while revising the introduction and discussion and have included the mentioned references.

Referee #3:

How E3 and E2 ligases cooperate and select each other is an important question that has been characterized for selected model systems. This manuscript analyzes the E2-E3 interplay for the yeast ERAD RING E3 ligases Hrd1 and Doa10 with Ubc6 and 7, and combines in vitro and cellular assays with NMR. It has previously been shown that Hrd1 works with Ubc7 in vivo while Doa10 uses Ubc6 and 7 for priming and elongation, respectively. In the present manuscript, the authors use kinetic analyses to conclude that Hrd1 uses Ubc7 for priming and that priming is the rate-limiting step in protein degradation directed by Hrd1 and Doa10. The abstract also claims that the study offers "a mechanistic framework to understand functional selection and coordination of E2 enzymes by RING E3 ligases in general". However, the paper falls short of showing evidence that the findings may be generalised. I feel that this point needs to be worked out more to add impact, in addition to the issues below, many of which are technical.

We thank the reviewer for the constructive criticism. We now point out in the discussion how the well-studied ERAD system offers unique properties to investigate differential E2 usage by RING E3s. We have restructured our arguments and discussion around the questions how a RING E3 pairs with different E2s for the different steps of poly-ubiquitylation and the role of E3-mediated E2 stimulation in these processes. We point out more that as about half of all yeast and human RING domains harbor non-canonical linchpin residues, our case-study of Doa10 with its non-canonical linchpin has broad relevance to the Ub field. In the revised discussion we outline published examples that appear consistent with our model of RING E3s that have properties tailored to their preferred E2 and/or that disfavor other E2s for certain reactions.

MAJOR CONCERNS:

Figure 1C. To compare the substrate ubiquitination activities of Hrd1 in the presence of UBC6 and 7, it is necessary to show these reactions on the same Western blot. The Western should be repeated and

quantified. As is, the data does not look convincing, since the right blot seems less exposed than the left one (see much darker shadow underneath the RNase band - what is this shadow anyway?).

We repeated the assay using a different α -RNase antibody, which increased the quality of the Westerns and allowed for quantification. New blots including quantifications of triplicate experiments are now shown in Fig. 2B, 4B, and 4D. The antibody also eliminated the shadow underneath the RNase band, which most likely arose from a cross-reaction of the former antibody with the RING domains as it was absent in “no E3” samples.

Figure 1D. The actual data appear to be missing. Those should be presented along with the quantification.

We added representative gels and decay curves to Fig. 2C and D to demonstrate the workflow of how we obtain the “fold stimulation” values for our presentation. Representative data for all other assays have been added to the supplements as well.

Figure 1 E, F: It remains unclear how and why "selected residues" were chosen. It may be more suitable to define some cut-off and include all residues within this range for KD analysis. This should also be done in the analyses in Figure 2S, because it may illustrate mutation induced differences in binding mode. If any, those should be discussed.

The procedure outlined by the reviewer was followed in the original submission (cut-off = residues in the 90th percentile of CSPs). However, in response to an issue raised by another reviewer, we have replaced the K_D analyses for all E2/E3 interactions with those for the more catalytically-relevant E2-SS-Ub/E3 interaction. NMR was not useful to determine these stronger K_D values, so we switched to ITC measurements.

Page 7: The authors state that "The linchpin mutations do not substantially alter E2/E3 binding affinity". To be able to conclude this, the NMR titration data should be fitted and KD values reported. In Figure S2G, it looks like the critical arginine mutation of the linchpin in Doa1 does have an effect, but without a fit, it is hard to say. For the fitting it would be important to use the same data range for mutants and WT (the same maximum concentration of ligand). To also account for possible changes in the binding mode, the authors should analyze all resonances within a certain cut-off (see comment above) and not 4 selected ones.

We agree with the reviewer’s assessment: In the titrations, we observed that the H94R and H94A mutations seem to have an effect on interaction between Doa10 and U7BR/Ubc7. Although we cannot do proper fitting from the data, we estimate the K_d to be only roughly 30% higher and lower for the Doa10(94R) and (94A) mutant, respectively. Importantly, these differences do not track with catalytic differences of the mutants compared to the wild-type: Doa10(94R) is about twice as active and (94A) as active as wild-type Doa10 in stimulating discharge from U7BR/Ubc7 (see Fig. 3A). This suggests that the difference in affinity for the E2/E3 interaction do not translate into altered catalytic activity.

As stated above, we have removed the NMR-based K_d analysis in favor of ITC data on the E2-SS-Ub/E3 interactions. We did not assess the interactions between E2-SS-Ub conjugates and E3 linchpin variants by ITC and have thus removed all claims and discussion about binding strength of the Doa10 linchpin mutants from the manuscript. As a side note: we obtained intensity-based NMR data for these interactions (data not shown), which suggest that the Doa10 linchpin variants bind the U7BR/Ubc7-SS-Ub to a similar degree as wild-type Doa10 and, importantly, in a manner that does not correlate with their corresponding enzymatic activities.

Figure 2A, B: The data from which the rates were derived should be shown in the supplements.

Representative gels and fluorescence scans from these assays have been added (see Appendix Fig. S1 – S9).

Page 8: The authors conclude "The simplest explanation for the disparity between the in vitro elongation kinetics and in vivo substrate degradation is that chain elongation is not rate-limiting to the process of protein degradation in yeast. This implies that the "minimal E3-stimulated elongation rates" observed in the in vitro assays with wild-type-RINGs (see Fig. 2B) are sufficient to guarantee effective protein degradation at both E3s under the conditions assayed in vivo." The first sentence makes sense. The second one seems overstated. There are many factors in the cell that may alter the rates compared to the in vitro situation...

We rephrased this part of the manuscript in a way that avoids overstatements and appreciates the complexity of the *in vivo* situation (pages 10-12). For illustration, we have added an overview of the processes involved in eventual ERAD substrate degradation (Fig. 3C).

Figure 2C: Protein degradation is monitored by pulse-chase experiments for the Hrd1 model substrate and by CHX decay assays for the Doa10 model substrate? The authors should give an explanation as to why two types of assays are necessary.

We regret not being able to show all degradation data using the same assay. Originally, we tried to monitor degradation of all substrates by CHX decay assays but experienced technical difficulties (detection/data quality needed for quantification) for the Hrd1 substrates. We therefore switched to pulse-chase experiments for these substrates. Sample preparation in pulse-chase assays is technically more variable (immunoprecipitation of substrate for every sample), so we limited its use as much as possible and kept the CHX analysis for Doa10 substrates.

Figure 3A: The errors look quite large, making it necessary to demonstrate significance.

We acknowledge that the errors in former Fig. 3A (now 4A) are quite large. Degradation assays involve a lot of work-steps, especially for pulse-chase analysis (see Materials & Methods). Thus, the variance for a given strain between replicates is high, resulting in large apparent errors, yet differences between strains within each replicate were consistent. To add confidence to our interpretations and conclusions, we followed two different substrates per ligase to assess if similar trends can be seen for either substrate (which is the case).

As suggested, we performed significance analyses for the results of Fig. 4A and for all *in vitro* data in the manuscript (see figure legends for details). Concerning Fig. 4A: we performed significance analyses of the difference in protein levels at the last time point. Thus, we report significant differences between Hrd1(400R) and $\Delta hrd1$, between Hrd1(400R) and Hrd1(400E) as well as between Hrd1(400H) and Hrd1(400E). Eventually, we decided against adding significance analyses for degradation assays to the figures to avoid overcrowding them.

Figure 3D: The authors state "the differences among linchpin mutations were far less pronounced than for Ubc7 (compare Figure 2A)" However, it seems that the difference between WT and R is about 3 fold and the same is seen in 2A.

We have rephrased this section of the manuscript to be more precise (pages 13-14). Our point here is not that Ubc6 is completely insensitive to the action of a linchpin, but rather that, due to its high basal activity, it is not utterly dependent on it for Ub transfer, unlike Ubc7. As the reviewer points out, results from discharge assays

with the Doa10 linchpin variant imply that Ubc6 can be stimulated when an optimal Arg linchpin is present. However, we point to results from discharge experiments with Hrd1 linchpin variants (Fig. EV2F+G) where the Arg linchpin does NOT result in increased stimulation of Ubc6 discharge. And while the Doa10(94R) variant showed hyper-activity in stimulating discharge from Ubc6, this behavior is not recapitulated for Ubc6-dependent *in vitro* ubiquitylation assays (Fig. 4D) nor does this variant lead to increased rates of substrate degradation in cells (Fig. 4A – right panel).

Figure 3E: To compare the linchpin dependencies of Doa10 with Ubc6 and 7, they should be on the same blot.

This has been done and in Fig. 4D. Additionally we added quantifications.

Figure 4A: This assay lacks a control (time point zero, minus ATP...) and +DTT samples to provide evidence for the thioester versus autoubiquitinated species (important for Ubc7~Ub in particular). Representative data for the reactions with nucleophile appear to be missing.

“Time point zero” as well as reducing controls with representative Coomassie-stained gels have been added to Fig. 2C, 5A and Appendix Fig. S1. We did not add “minus ATP” controls, since the discharge reactions were not started by addition of ATP but of nucleophile. Instead, we added controls showing the charging reactions before addition of ATP (see Fig. 2C – first lanes).

Figure 5: Representative raw data appear to be missing. Likewise controls, e.g. experiments illustrating the identity of the yeast strains.

Autoradiographs and blots have been added (see Appendix Fig. S12-S15 and S19). All strains have been verified (compiled in Appendix Fig. S20). We addressed protein levels where possible. As we do not have an antibody targeting Ssh1, we checked the deletion of *SSH1* by PCR on genomic DNA. Additionally, we amplified the genomic locus of *DOA10* or *CUE1* in strains harboring Doa10 variants or Cue1(RGA). Sequencing results of all tested strains can be provided if desired.

Figure S4B/D: The RGA mutation makes an obvious difference here. However, the authors state "the Cue1(RGA) mutation that is so detrimental to *in vitro* elongation rates had virtually no effect on substrate degradation kinetics for either E3 (Fig. 2C and Fig. S4A+B - compare solid vs. dashed traces in each panel)" Also, the authors state that "virtually no difference in the degradation rates of Doa10 substrates was observed (Fig. 3C and Fig. S4D). However, there is a clear differences between the mutants in S4D. The authors should explain their interpretation and/or make it more precise.

We have addressed these points by rewriting the mentioned parts of the manuscript to be more precise and to comment on these differences (pages 10-11).

Figure S5: Why is the signal in the top blot so much stronger on the left than on the right? It makes it hard to interpret.

We removed former Fig. S5 from the manuscript, as we reran all *in vitro* ubiquitylation assays with a different RNase antibody that does not lead to the artifact mentioned by the reviewer (see Fig. 2B, 4B and 4D).

MINOR CONCERNS:

Page 4: The authors state "There are limited examples of RING E3s that are confirmed to collaborate with a priming and an elongation E2 to catalyze attachment of poly-Ub chains to substrates (Parker and Ulrich, 2009; Rodrigo-Brenni and Morgan, 2007; Weber et al., 2016; Wu et al., 2010). Therefore, understanding of the specific role of E3 ligases and their interplay with E2 enzymes during the steps of poly-ubiquitylation is still rather limited. "

This overlooks landmark work by Schulman, Rape and others that has provided exquisite detail of the roles of priming and elongating E2s in the context of the APC. This work should be included in the introduction. When introducing the closed conformation (page 3), the authors should cite Deshaies and Rape (Saha et al., Mol Cell 2011 and Wickliffe et al., Cell 2011).

The introduction has been rewritten and relevant publications have been added.

Page 5: The authors show by NMR that Hrd1 has significantly different affinities for Ubc6 and 7, with the latter interaction only in the millimolar range. Doa10 binds both E2s only in the millimolar range (yet works with both E2s). This suggests, indeed, that Hrd1 may use affinity to discriminate the E2s. However, it also shows that the in vitro affinities only partially reflect what is happening in the cell (since Doa10 works with both E2s, despite both being in the millimolar affinity range in vitro). The authors should quote the KD values in the text and comment on this point, since the situation This is more complex than the current interpretation indicates.

We appreciate that the situation in the cell is more complex and have added a paragraph (pages 6) to explicitly state that our data refers only to the soluble parts of the ERAD ubiquitin enzymes. Moreover, we show in the revised manuscript that the E2~Ub conjugates have higher affinities (μM Kd values) to the RINGs than the corresponding free E2s and acknowledge that tethering of all components to the ER membrane may lead to higher local concentrations.

Figure 1A: The table needs some streamlining. It is meant to list the functions of each protein, but lists tagged substrates and general properties.

The table has been streamlined (Fig. 1D).

Figure S6C: One of the blue data needs recoloring, since they are indistinguishable in the legend.

This has been corrected and is now found in Fig. 6B.

Figure S1D: The highlighted residues should be labeled on the structures.

We have removed this analysis from the manuscript due to the removal of the NMR-based E2/E3 interaction study (see comments above).

Figure S1G-H: Aren't there NMR assignments of yeast Hrd1 that could be transferred here?

NMR assignments for yeast Hrd1 are available on BMRB, however transfer of these assignments to the Hrd1 construct we used in this study has not proven to be trivial. As we have removed the NMR-based Kd analysis in our revision, we did not pursue this point further.

- Bazirgan, O.A., Hampton, R.Y., 2008. Cue1p is an activator of Ubc7p E2 activity in vitro and in vivo. *J. Biol. Chem.* 283, 12797–12810. doi:10.1074/jbc.M801122200
- Biederer, T., Volkwein, C., Sommer, T., 1997. Role of Cue1p in ubiquitination and degradation at the ER surface. *Science* 278, 1806–1809. doi:10.1126/science.278.5344.1806
- Choi, Y.-S., Lee, Y.-J., Lee, S.-Y., Shi, L., Ha, J.-H., Cheong, H.-K., Cheong, C., Cohen, R.E., Ryu, K.-S., 2015. Differential Ubiquitin Binding by the Acidic Loops of Ube2g1 and Ube2r1 Enzymes Distinguishes Their Lys-48-ubiquitylation Activities. *Journal of Biological Chemistry* 290, 2251–2263. doi:10.1016/j.jmb.2011.01.047
- Dove, K.K., Stieglitz, B., Duncan, E.D., Rittinger, K., Klevit, R.E., 2016. Molecular insights into RBR E3 ligase ubiquitin transfer mechanisms. *EMBO Rep.* doi:10.15252/embr.201642641
- Kostova, Z., Mariano, J., Scholz, S., Koenig, C., Weissman, A.M., 2009. A Ubc7p-binding domain in Cue1p activates ER-associated protein degradation. *Journal of Cell Science* 122, 1374–1381. doi:10.1242/jcs.044255
- Metzger, M.B., Liang, Y.-H., Das, R., Mariano, J., Li, S., Li, J., Kostova, Z., Byrd, R.A., Ji, X., Weissman, A.M., 2013. A structurally unique E2-binding domain activates ubiquitination by the ERAD E2, Ubc7p, through multiple mechanisms. *Molecular Cell* 50, 516–527. doi:10.1016/j.molcel.2013.04.004
- Pruneda, J.N., Littlefield, P.J., Soss, S.E., Nordquist, K.A., Chazin, W.J., Brzovic, P.S., Klevit, R.E., 2012. Structure of an E3:E2~Ub complex reveals an allosteric mechanism shared among RING/U-box ligases. *Molecular Cell* 47, 933–942. doi:10.1016/j.molcel.2012.07.001
- Pruneda, J.N., Stoll, K.E., Bolton, L.J., Brzovic, P.S., Klevit, R.E., 2011. Ubiquitin in Motion: Structural Studies of the Ubiquitin-Conjugating Enzyme-Ubiquitin Conjugate. *Biochemistry* 50, 1624–1633. doi:10.1021/bi101913m
- Wickliffe, K.E., Lorenz, S., Wemmer, D.E., Kuriyan, J., Rape, M., 2011. The mechanism of linkage-specific ubiquitin chain elongation by a single-subunit E2. *Cell* 144, 769–781. doi:10.1016/j.cell.2011.01.035

Thank you for submitting your revised manuscript for our consideration. It has now been assessed once more by reviewers 2 and 3, and I am happy to inform you that both consider the study significantly improved in response to the original comments. We shall therefore proceed with publication in The EMBO Journal, following incorporation of a number of editorial issues remaining:

REFEREE REPORTS

Referee #2:

The authors have addressed my initial concerns and have improved their manuscript. I support publication.

Referee #3:

The authors have addressed my points comprehensively. Too bad that the NMR data were removed, but I feel the story has much improved overall.

YOU MUST COMPLETE ALL CELLS WITH A PINK BACKGROUND ↓
PLEASE NOTE THAT THIS CHECKLIST WILL BE PUBLISHED ALONGSIDE YOUR PAPER

Corresponding Author Name: Rachel E. Klevit
Journal Submitted to: EMBO Journal
Manuscript Number: EMBOJ-2020-104863